# FAMOUS version xotzt (FAMOUS-ice): a GCM capable of energy- and water- conserving coupling to an ice sheet model

Robin S. Smith[1,2], Steve George[1,2], and Jonathan M. Gregory[1,2,3]

[1]NCAS-Climate, University of Reading, U.K.
[2]Meteorology Department, University of Reading, U.K.
[3]Met Office Hadley Centre, Exeter, U.K.

**Correspondence:** Robin S. Smith
(robin.smith@ncas.ac.uk)

**Abstract.**

The physical interactions between ice sheets and their surroundings are major factors in determining the state of the climate system, yet many current Earth System models omit them entirely or approximate them in a heavily parameterised manner. In this work we have improved the snow and ice sheet surface physics in the FAMOUS climate model, with the aim of improving

the representation of polar climate and implementing a bidirectional coupling to the Glimmer dynamic ice sheet model using the water and energy fluxes calculated by FAMOUS. FAMOUS and Glimmer are both low resolution, computationally affordable models used for multi-millennial simulations. Glaciated surfaces in the new FAMOUS-ice are modelled using a multi-layer snow scheme capable of simulating compaction of firn and the percolation and refreezing of surface melt. The low horizontal resolution of FAMOUS compared to Glimmer is mitigated by implementing this snow model on sub-gridscale tiles that rep-

resent different elevations on the ice sheet within each FAMOUS grid-box. We show that with this approach FAMOUS-ice can simulate relevant physical processes on the surface of the modern Greenland ice sheet well compared to higher resolution climate models, and that the ice sheet state in the coupled FAMOUS-ice-Glimmer system does not drift unacceptably. FAMOUS-ice coupled to Glimmer is thus a useful tool for modelling the physics and co-evolution of climate and grounded ice sheets on centennial and millennial timescales, with applications to scientific questions relevant to both paleoclimate and

future sea level rise.

# 1 Introduction

Continental-scale ice sheets are one of the primary components in the Earth's climate system. The climatic influences that result as they grow and shrink are key features of the global-scale glacial cycles of the last million years. Their waxing and waning volume causes sea-level variations of over 100m in amplitude over these cycles (Spratt and Lisiecki, 2016) through their barystatic contribution (the global-mean effect of changing the mass of the ocean), by altering the gravitational field and rotation of the Earth and by deforming its solid surface (sea level concepts and terminology are reviewed by Gregory et al., 2019). Ice sheet mass loss accounts for around a third of the present rate of global mean sea-level rise; their contribution is expected to increase and likely eventually dominate sea-level change in the coming decades and centuries (Oppenheimer et al., 2019).

The role of ice sheets in the climate system involves strong bidirectional interactions with both the atmosphere and ocean. As ice sheets evolve they substantially modify the surface radiation balance and temperature by changing the albedo and altitude of the Earth's surface, the supply of fresh water to the ocean, and the circulations of both the atmosphere and the ocean (e.g. Golledge et al., 2019). The diversity and strength of these feedbacks means that ice sheets should properly be modelled as physically coupled, interactive components in climate models when addressing many scientific questions related to the evolution of the Earth system. Such questions include projections of the impacts of future climate change on multi-decadal timescales and longer due to the increasing influence of ice sheets on the global mean sea-level budget (e.g. Edwards et al., 2014; Nowicki et al., 2016). However, current understanding of the co-evolving physics of our climate and ice sheets is limited in detail. This hinders our ability to explain major features of climate change in the past and make projections about what may happen in the future.

There are numerous challenges in successfully modelling the physics of ice sheet–climate interactions in the atmosphere-ocean general circulation models (AOGCM) that are commonly used for comprehensive climate studies. Many ice sheet-relevant processes have characteristic length scales of kilometres or less (e.g. gradients of precipitation and surface melt on the sloping margins of Greenland, or ocean melt at the grounding line of Antarctica's floating ice shelves). This is much less than the grid-box size of tens or hundreds of kilometres in contemporary AOGCMs (e.g. Sellar et al., 2019; Danabasoglu et al., 2020). Many of the ice sheet features that depend on these small scale processes can, however, evolve on millennial timescales, implying climate simulation lengths that are computationally unaffordable with complex AOGCMs. The technical structure of most AOGCMs also makes them ill-suited to changing the boundaries of the land or ocean domain as they run. This issue is a particular problem where the evolution of the ice involves the collapse of an ice shelf or a change in global sea-levels. Furthermore, many of the physical processes that are key to modelling the surface physics of polar regions are not well captured by models that are intended for global use, such as stable polar boundary layers, katabatic winds or multi-year snow packs (Connolley and Bracegirdle, 2007).

For all these reasons, the majority of AOGCMs do not include interactive ice sheets, and omit much of the physics required to directly model boundary conditions for ice sheet models. For their part, ice sheet modellers have developed empirical parameterisations to translate the climate fields that AOGCMs do provide into usable boundary conditions (e.g. Reeh, 1991),

including methods of adapting the climate data for a changing ice sheet geometry despite the AOGCM climates having been simulated with a fixed ice sheet (e.g. Edwards et al., 2014; Goelzer et al., 2020). Until recently, most of the existing coupled climate-ice sheet models relied on such parameterisations rather than having a direct physical coupling between the climate and the ice (e.g. Bonelli et al., 2009; Gregory et al., 2012; Roche et al., 2014). In general, these coupled models use climate models of reduced physical complexity and resolution that can practically run millennial-scale climate simulations to match the timescales of ice sheet evolution.

Recent advances in the fields of both climate and ice sheet modelling are starting to change this situation. More powerful computers and increased interest in modelling the physics of the "Earth System" beyond atmospheric and ocean physics have led to improvements in the representation of polar processes in the components of AOGCMs (e.g. Punge et al., 2012; Mathiot et al., 2017). Motivated mainly by the need to project sea-level rise over the coming centuries, a number of models have been developed to include more sophisticated surface schemes for ice sheet regions, and some have coupled these schemes to ice sheet models (ISM) (e.g. Sellar et al., 2019; Danabasoglu et al., 2020).

The mismatch in spatial resolution between the grids used for key atmosphere and ice sheet processes has been addressed either through explicitly modelling the local atmosphere at much higher resolution in a regional or nested model (van Kampenhout et al., 2019), or by calculating sub-gridscale surface fluxes and mass balance terms for portions of atmospheric grid-boxes (Ganopolski et al., 2010; Vizcaíno et al., 2013; Ziemen et al., 2014; Sellar et al., 2019). Using sub-gridscale methods is often more computationally affordable than increasing the explicit resolution of the atmosphere model, so allows much longer coupled climate-ice sheet simulations to be carried out. Many of the quantities required for these sub-gridscale calculations are well correlated with surface temperature, and in regions with significant topographic gradients—like the margins of an ice sheet—are also a strong function of altitude. Several models now exist that treat each land surface grid-box in ice sheet areas as a collection of sub-gridscale tiles, each representing a different elevation range within the grid-box (e.g. Vizcaíno et al., 2013; Sellar et al., 2019). The tiles are aggregated into a grid-box average for communication with the atmosphere, but used individually when required to provide finer-grained information to an ice sheet model.

FAMOUS is a relatively coarse resolution AOGCM that has been used successfully in a wide range of climate studies (e.g. Smith and Gregory, 2012; Hawkins et al., 2016; Joshi et al., 2017; Dentith et al., 2019) and is computationally cheap enough for multi-millennial simulations. In previous coupled climate-ice sheet studies with FAMOUS (Gregory et al., 2012; Roberts et al., 2014), the coupling was achieved through a simple positive-degree-day parameterisation and it was later found that the long-term response of the ice was very sensitive to ill-constrained empirical parameters within this scheme. Building on FAMOUS's climate simulation and a framework for sub-gridscale modelling within its land surface scheme, we decided to improve FAMOUS's representation of ice sheet surface physics and to directly couple water and energy fluxes between FAMOUS and the ice sheet model.

In this paper we describe modifications that we have made to the physics and tiling in the land surface of FAMOUS to produce a configuration called FAMOUS-ice, and we show that the results are scientifically useful within the context of coupling to a model of the Greenland ice sheet (GrIS). We do not attempt to model any ocean-ice sheet interaction in this work, as the fjords and marine-terminating glaciers of Greenland are not resolved on the FAMOUS grid. We focus primarily

on technical aspects of our work, including the scheme by which FAMOUS-ice has been directly coupled to the Glimmer ice sheet model. Simulation results shown in this paper are illustrative, intended to demonstrate the capabilities of the coupling techniques in our model and to give additional background context of typical physical behaviour of FAMOUS-ice to aid interpretation of future studies, rather than for in-depth analysis or evaluation of the large-scale climate of the FAMOUS atmosphere (see e.g. Smith, 2012) or the general behaviour of the Glimmer ice sheet (see e.g. Rutt et al., 2009). We will not address issues that significantly affect the setup of simulations aimed at specific questions of coupled climate-ice modelling (for instance coupled spinup and initialisation). A scientific application of FAMOUS-ice, looking at the future stability of the Greenland ice sheet, is published separately (Gregory et al., 2020).

## 2 Standard FAMOUS

FAMOUS (Smith et al., 2008; Williams et al., 2013) is a configuration of version 4.5 of the UK Met Office Unified Model (UM4.5, Gordon et al., 2000). The atmosphere in FAMOUS has a horizontal resolution of 7.5°x5°, with 11 vertical levels. The development described here is based on the most recently released configuration of FAMOUS, xfhcu (Williams et al., 2013). In the present work, the ocean model is replaced by prescribed sea surface boundary conditions taken from a set of higher resolution AOGCM simulations. This allows us to evaluate the performance of the ice sheet surface mass balance scheme with minimal bias in the wider simulated climate.

FAMOUS xfhcu uses the MOSES2.2 land surface model (Essery et al., 2003), an early version of the JULES land surface scheme (Best et al., 2011) used in current UM configurations (e.g. HadGEM3-GC3 and the UK Earth System model (UKESM1) Sellar et al., 2019; Williams et al., 2018). MOSES2.2 shares the same grid resolution as the overlying atmosphere model, and can represent heterogeneity in land surface characteristics by using a set of sub-gridscale tiles in each grid-box, each of which simulates a different type of surface. Each tile covers a particular fraction of the area of its grid-box, but is not explicitly geographically delimited. Effects at the grid-box-scale are given by an average across the set of tiles in that box, weighted by the fraction of the area of the grid-box that each tile covers. Tiles in MOSES are generally used to represent different types of surface vegetation. FAMOUS xfhcu, like most UM configurations with tiles to date, uses 9 different surface types, one of which represents permanently glaciated surfaces. The underlying soil in xfhcu is only represented by entire grid-boxes; all surface tiles in a grid-box share a common sub-surface. Because the glaciated ice surface tile requires its own distinct ice sub-surface, land ice cannot share a grid-box with any other surface tile, and entire grid-boxes must be either entirely glaciated or ice-free. This means that the spatial representation of the edge of an ice sheet in MOSES2.2 is usually very blocky in a low resolution UM configuration like FAMOUS.

MOSES2.2 as described in Essery et al. (2003) has a "zero-layer" snow model, which represents only the bulk properties of the snow. This model assumes that snow has a fixed density (250 kg/m$^3$) and, if deep enough, can insulate the surface from the atmosphere. Snow surface albedo is calculated in four radiation bands (direct and diffuse radiation in visible and infrared bands) from the grain-size of the snow, which may evolve with time and surface temperature. The underlying ice tile has a prescribed albedo of 0.75 in all bands. The performance of the MOSES albedo scheme and the lack of internal

snow pack refreezing of surface melt were identified as particular shortcomings for the calculation of ice sheet surface mass balance (SMB) in previous work with the UM (Rae et al., 2012). To improve the representation of SMB in glaciated areas in FAMOUS-ice we have significantly modified the capabilities and behaviour of the tiles, snow pack model and snow and ice surface albedos. The body of the snow scheme we describe in the following has largely been backported from version 3 of JULES, with further modifications to both the snow and the tiling system that have been done largely in parallel with work in more recent versions of JULES and UKESM.

## 3  Tiles for fractional ice extent and sub-gridscale elevation

In FAMOUS-ice, we have modified the MOSES2.2 tiling scheme to allow each of the 9 basic surface types to exist at multiple fixed-height elevations within a grid-box. Thus, calculating surface exchange for each tile separately, we simultaneously simulate the different conditions that would be seen on ice surfaces at, for example, sea-level, 100m and 500m, with the same large-scale atmospheric column above the boundary layer for each. In this paper we have configured these height-dependent tiles only on Greenland to minimise computational cost, although they could be used globally in principle. The resulting three-dimensional (longitude, latitude, height) arrays allow the climate model output for the GrIS to be interpolated in both horizontal and vertical dimensions onto a finer resolution ice sheet model grid, while the atmosphere uses the grid-box mean of the tiles, weighted by their area fractions. All the fine-grid climate information seen by the ice sheet model is thus consistent with the large-scale average in the climate model as it evolves.

In FAMOUS-ice, the ice and soil sub-surface models are allowed to co-exist, so that each grid-box may contain any combination of ice and non-ice surface tiles at the designated altitudes. This allows the atmosphere to see a sub-gridscale, fractional representation of area at the edge of the ice sheet, rather than binary blocks of entire ice/not-ice grid-boxes. The soil and ice sub-surface models still operate at the grid-box-scale and are not independent for each surface tile. This means that tiles at all elevations are coupled to a common sub-surface layer. Section 4 describes how tiles at different elevations communicate with their sub-surface in such as way as to minimise the spurious flow of information between them.

Figure 1 compares the frequency distributions of height surfaces for a sample GrIS topography in FAMOUS-ice, using the standard FAMOUS grid-box mean orography and two choices of distributions of elevation tiles. The finer-scale detail generated for the ice sheet model comes from an increase in the total number of surface tiles simulated and a closer approximation to the true frequency distribution of surfaces with height as the number of elevation classes used within each grid-box increases. Increasing the number of tiles used implies both computational overhead and an increase in the memory required for I/O, which can be slow in FAMOUS, so it is wise to not to use more tiles than really needed for a given application. In practice, we find the use of 10 tiles, with the same vertical distribution as is used in CESM1 (Vizcaíno et al., 2013) with boundaries at [0, 200, 400, 700, 1000, 1300, 1600, 2000, 2500, 3000, 10000] metres, is a good compromise. This increases the number of surfaces calculated on the GrIS in FAMOUS by a factor of 10 compared to using the grid-box mean, and adds many more surfaces at low elevations which is important for simulating higher ablation areas.

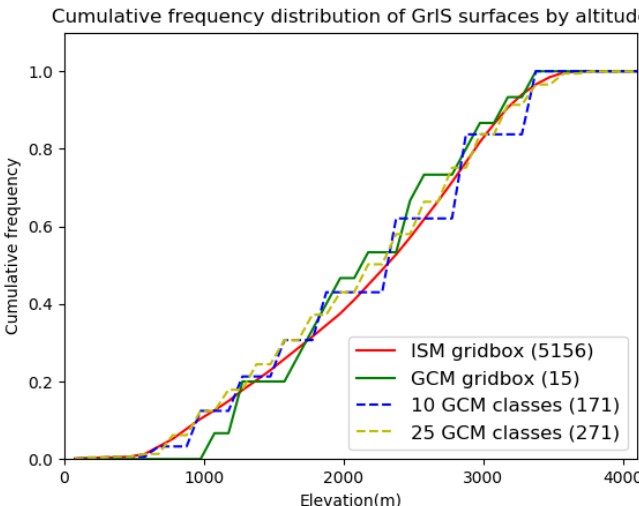

**Figure 1.** The distribution and number of individual surfaces modelled on the Greenland ice sheet on different grids. Red: a 20km ice sheet model; green: normal FAMOUS grid-boxes; blue dashed: FAMOUS-ice with 10 sub-gridscale elevation classes; yellow dashed: FAMOUS-ice with 25 elevation classes. The total number of surfaces modelled for Greenland in each representation is given in brackets.

## 4  Downscaling climate to the tiles

In order to reproduce differences in snow pack and ice evolution on different elevation tiles in a grid-box—for example snow/ice mass loss (ablation) at lower, warmer altitudes and net accumulation above the equilibrium line—each tile must experience atmosphere variables adjusted to the particular altitude that it represents. This requires a downscaling procedure that is com-
putationally fast enough to be applied timestep-by-timestep in the atmosphere model, and which conserves the grid-box-mean of the tiles and the physical relationships between the variables being scaled. We require the scheme to provide good results for local altitude adjustments only over a specific ice sheet; the downscaling does not need to work globally. We focus further on downscaling only those fields which are required to produce significant differences in surface mass balance at different altitudes in a single grid-box.
Global, fixed lapse rates are often used to downscale climate model variables onto higher resolution ice sheet surfaces, especially temperature for use in degree-day or other temperature-index schemes (e.g. Roche et al., 2014; Vizcaíno et al., 2013).

In preliminary work we tried more sophisticated methods, sampling the local atmospheric lapse rates for each grid-box and attempting to maintain net zero averages in the sum of the adjustments for the tiles, but in a low resolution model like
FAMOUS where 11 vertical layers must account for the whole depth of the atmosphere, the free-atmosphere lapse rate this calculation produces is often not a good approximation for how near-surface temperatures should vary with height. Ultimately, we found the best results for downscaling near-surface air temperature and downwelling longwave radiation onto the elevation

tiles came from prescribing spatially constant lapse rates. Optimal values for the lapse rates of 6 K/km (for temperature) and 3.6 $W/m^2/K$ for longwave radiation were found by comparing lower-atmospheric conditions in FAMOUS-ice downscaled to elevation tiles over the GrIS with output from the MAR regional model (Fettweis et al., 2013). This empirically tuned approach is open to question if the ice sheet (in a coupled model) is allowed to evolve to a significantly different state from that the lapse

rates were calibrated for. However, we gain confidence from the results of our model in a climate-change experiment in which the Greenland ice sheet is reduced to a small fraction of its present size; we find that the ice sheet area-mean summer-mean surface air temperature change as a function of change in surface altitude is close to our chosen lapse rate (Gregory et al., 2020).

Specific humidity is downscaled using a lapse rate derived from the local atmospheric lapse rates for each grid-box. Precipi-

tation is not adjusted for the tiles, either in terms of the magnitude or partitioning between snow and rain, because of complications in robustly maintaining consistent vertical energy budgets in the atmospheric column when changing the amount or phase of moisture. Each tile in a grid-box receives the same amount and phase of precipitation, calculated with reference to the mean grid-box orographic height. Downwelling shortwave radiation and surface momentum fluxes are also not downscaled because in FAMOUS-ice they are found to vary negligibly across the elevation range used for the tiles.

All tiles within a grid-box share the same soil or ice sub-surface, regardless of their elevation, so it is also necessary to adjust the bottom boundary layer temperature of each elevation tile to limit the potential for unrealistic heat flows between the tiles via their shared sub-surface. For example, a polar latitude, cold ice tile with a temperature of 250K at the bottom of the snow pack should not see the same sub-surface boundary temperature as a tile at sea level in the same grid-box where conditions may be causing the snow to melt. A sub-surface temperature lapse rate of -1 K/km was used. Although not formally tuned, this

value is in line with the variation of sub-surface temperature with mean orographic height under GrIS snow packs simulated in this run for sample transects across the ice sheet, and experimentation with the ice sub-surface formulation during model development suggests that varying the sub-surface lapse rate would not be expected to have a major effect on our results. We do not to adjust moisture availability to the elevated tile from the soil sub-surface.

Example results show how surface exchange fluxes vary with height on the ice sheet in FAMOUS-ice compared to those from

regional climate model (RCM) simulations of the GrIS in MAR (Fettweis et al., 2013) and RACMO (Noël et al., 2018) (figure 2. Supplementary figures A1-A5 show spatial variation of these quantities across the ice sheet). Average vertical gradients of all quantities are of reasonable magnitude and the same sign as in the RCMs, improving on the elevation class implementation analysed by Sellevold et al. (2019) (figure A6).

As in this example, FAMOUS-ice often simulates a low bias in downwelling shortwave over the GrIS (figure 2), likely due to

excessive cloud over the ice sheet in summer. The spatial distribution of downwelling shortwave over the GrIS in MAR forced by the 1980-1999 MIROC5 background climate (Watanabe et al., 2010) used in this example is also very different from that in FAMOUS-ice (figure A1), suggesting that the discrepancy is not simply a difference in local cloud amounts but also a difference in the large scale circulation over the GrIS between the models. Compared to MAR there is more downwelling longwave radiation at the surface which is line with the excessive cloud hypothesis, although downwelling longwave is also influenced

by the temperature of air advected over the ice sheet. Summer surface air temperatures show a similarly orographically-

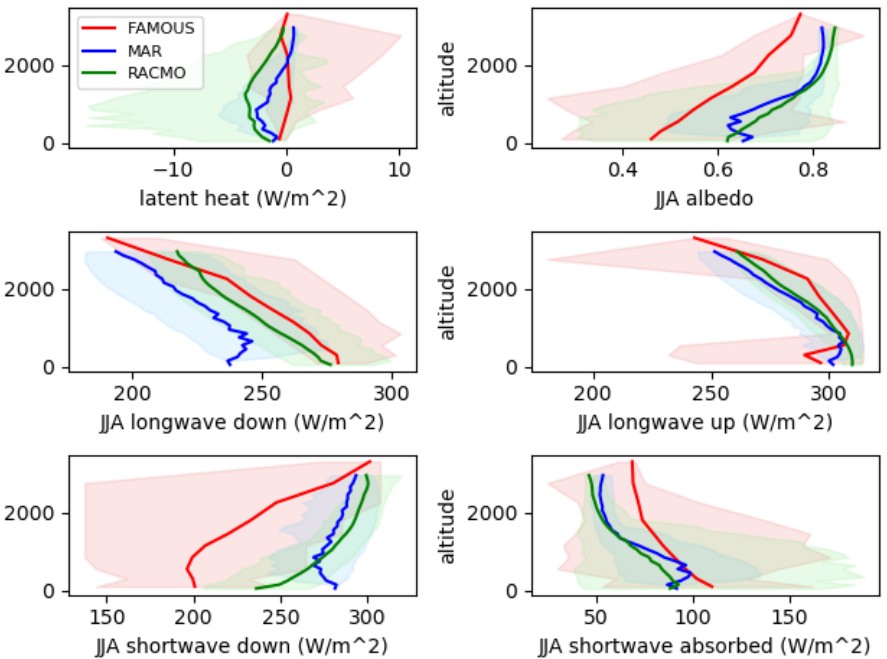

**Figure 2.** Profiles of surface exchange characteristics taken from every surface modelled on the Greenland ice sheet binned by surface elevation. Red: FAMOUS-ice; blue: MAR (1980-1999) (Fettweis et al., 2013); green RACMO (1980-1999) (Noël et al., 2018). FAMOUS-ice and MAR are forced by the MIROC5 climate (1980-1999) (Watanabe et al., 2010), RACMO by ERA-interim (1980-1999) (Dee et al., 2011). Latent heat is an annual average, negative values imply sublimation. Shading represents the full range of values found in each time-averaged elevation bin.

controlled pattern in FAMOUS and MAR, although FAMOUS does not give the same spatial gradient of the change of surface temperature with elevation seen in the RCM and is consequently warmer at higher altitude (figure A7). Since melting of snow is calculated by an energy balance model in FAMOUS-ice, surface air temperature does not have a direct control over the amount of melt it simulates. Summer surface albedo in FAMOUS-ice is clearly smaller than both RCMs at most heights on the

5   ice sheet - this will be discussed in section 6. The downwelling shortwave and albedo biases act in opposite directions, and the resultant amount of solar radiation absorbed by the ice sheet is more in line with the RCM simulations. Area integrated fluxes of latent and sensible heat (figure A5) are minimal over the GrIS in FAMOUS-ice, with low levels of sublimation across most of the ice sheet apart from an area of deposition, strongest in summer, in the south east. This is likely due to the specified air temperature lapse rate being inappropriate for this region. Shannon et al. (2019) used a very similar tile downscaling scheme

10   to compute glacier SMB in JULES forced by climate model data and found that they needed to apply site-specific empirical factors to enhance wind speed to obtain turbulent heat fluxes of a reasonable magnitude. Given the resolution of the FAMOUS-ice atmosphere our wind speeds are almost certainly too low, and we do not attempt to scale them by tile elevation. Turbulent flux magnitudes are significantly smaller than radiative fluxes on average in the RCMs and FAMOUS-ice.

## 5    Multi-layer snow-ice scheme

In FAMOUS-ice, we have replaced the MOSES zero-layer snow scheme with the multi-layer snow scheme now available in JULES. This model simulates vertical gradients of temperature and density within the snow pack, mechanical compaction, and the internal percolation, retention and refreezing of melt. The scheme was originally devised to simulate seasonal snow packs, so we have modified it for perennial snow, firn and solid ice. We use it in FAMOUS-ice to model the whole upper portion of the ice sheet that is relevant for SMB and climate.

The multi-layer scheme specifies a maximum number of layers and a maximum depth for each layer, aside from the lowest one. The bottom layer is allowed to thicken as much as is required to allow the total accumulated snow mass to be accounted for. Snow falling on a bare surface will initially accumulate in the first layer, until that layer's maximum depth is reached. Further snowfall results in mass originally in the first layer being transferred into a new layer underneath, so that the first layer does not exceed its maximum depth. As more snow accumulates this process continues, so that the top layers always contain the most recently fallen snow and new layers below are created and snow moved down into them as required (if permitted by the criterion for the maximum number of layers) to maintain fixed depths for the overlying layers. Loss of mass from the surface reverses this relayering process, and snow is moved back up to ensure that the layers nearest the surface are always full. Heat diffuses through the snow pack depending on energy fluxes at the top and bottom boundaries, snow grain-size grows as a function of time and layer temperature, and underlying layers are gradually compacted from the weight of snow above them. These properties are also relayered when mass is moved between levels, so that over the course of a year surface snow may be buried by new snowfall, compacted, then brought back to the surface with higher density and grain-size in the melt season. Both solid ice and any water retained in pore space count toward the density of each snow layer, which is recalculated every timestep. In FAMOUS-ice fresh snow has a density of $250\mathrm{kg/m}^2$, and it is not uncommon for compacted firn with a density above $800\mathrm{kg/m}^2$ to be revealed at the surface of the GrIS at low elevations during the summer.

Snow pack mass is not capped in this scheme, and without a mechanism to move mass laterally, snow depth may increase indefinitely in areas of the ice sheet where annual accumulation is greater than ablation. This has no direct effect on the physics of surface exchange with the atmosphere, nor is snow depth taken account of in the orographic height in FAMOUS. Likewise, without a mechanism to supply mass to areas that experience net ablation, regions of the ice sheet below the equilibrium line will eventually run out of snow to melt and runoff fluxes in the climate model will be distorted. In FAMOUS-ice, coupling to a dynamic ISM provides a mechanism to move mass from the accumulation zone to the ablation zone. As noted in section 8.1, by design we ensure that all ice tiles in FAMOUS-ice maintain sufficient snow such that it could not possibly all be melted between ice sheet coupling intervals, but the initial depth we choose for this is somewhat arbitrary. Here we initialise to a total depth 100m of snow on all ice tiles.

The grain-size of snow crystals is a primary factor in determining its optical properties, and thus the albedo of snow-covered surfaces. Snow grain-size increases from that of pure, fresh snow as the crystals deform and merge under the influence of the temperature, moisture content and pressure of their environment in complex ways (Colbeck, 1982). In the JULES snow model, grain-size evolves over time from a minimum, fresh snow value of 50 $\mu$m to a maximum of 2000 $\mu$m at a rate that is dependent

solely on the layer temperature and the current grain-size. The dependence of snow albedo on grain-size will be described in section 6.

The pore space available for melt-water retention in the original JULES multi-layer snow scheme scales with the depth of the snow pack. Surface melt can percolate downward through the snow, being retained where there is unfilled pore space in a
layer (and refrozen if the layer is cold enough) or passed on to the layer beneath if there is not. Where snow accumulates year on year and has become many metres deep, scaling the available pore space solely with depth leads to the unrealistic retention of all the surface melt. To counter this effect we additionally make pore space availability a function of the density of snow in each layer. Once the layer density increases above a threshold, the potential pore space amount is reduced linearly, reaching zero when the density reaches that of solid ice. Upon encountering an ice-density layer, any percolating melt is ejected from
the snow pack as runoff. Melt-water leaving the bottom snow layer is also treated as runoff by the land surface scheme. The pore space available for holding water $P$ in layer $k$ is formulated as:

$$
P_k = \begin{cases} P_{\mathrm{max},k} & \rho_{\mathrm{snow},k} < 450 \text{ kg/m}^3 \\ P_{\mathrm{max},k} \times \frac{850 - \rho_{\mathrm{snow},k}}{450} & 450 < \rho_{\mathrm{snow},k} < 850 \text{ kg/m}^3 \\ 0 & \rho_{\mathrm{snow},k} > 850 \text{ kg/m}^3 \end{cases}
$$

$$
P_{\mathrm{max},k} = S \times \rho_{\mathrm{water}} \times \mathrm{dz}_k
$$

where: $S$ is a liquid-retention parameter = 0.05, $\mathrm{dz}_k$ is the thickness of snow layer $k$, $\rho_{\mathrm{water}}$ is the density of water = 1000
kg/m$^3$ and $\rho_{\mathrm{snow},k}$ is the density of snow layer $k$.

The most recently-released UK Met Office AOGCM, HadGEM3-GC3 (Williams et al., 2018), is their first to use the JULES multi-layer snow scheme. HadGEM3-GC3 is configured with 3 snow layers, with interfaces at 0.04m and 0.16m. This configuration has been shown to simulate seasonal snow well. In this 3 layer configuration, the mass of snow in the two thin upper layers can never be enough to compact the lowest layer significantly. The density of the lowest layer thus remains permanently
near its initial low value, however much snow accumulates and however long it stays there. A perennial, deep snow pack in this configuration thus does not become compacted and retains all surface melt in the pore space of its thick lowest layer, producing no runoff. For FAMOUS-ice we retained three thin top layers, and added additional layers beneath to be used for perennial snow in glaciated areas. We chose to use 10 snow layers in total, with interfaces at 0.1, 0.35, 1.0, 3.0, 6.0, 10.0, 20.0, 40.0 and 70.0 metres. On glaciated tiles, we initialise the snow pack with 100 metres of ice-density snow, which is unable to
hold melt-water. The upper layers contain sufficient mass to cause compaction to occur in lower layers. The seasonal signal of temperature variation is resolved in the uppermost layers, while the lowest layers are insulated from variations on sub-annual timescales. This configuration of the multi-layer snow scheme provides a representation of snow physics both for glaciated areas and for seasonal snow in other areas.

## 6   Snow and ice albedo

The basic FAMOUS radiation scheme divides the incoming shortwave into two spectral bands, one for the visible and one for near-infrared, with distinct albedos. For tiles covered by less-dense snow, these albedos are based on the surface grain-size. Where the snow pack has completely melted away and the underlying sub-surface is exposed to the atmosphere, albedos for each band are instead prescribed according to sub-surface type; for the glaciated portion of the grid-box, this is bare ice, which has lower albedo than snow. The albedo over the GrIS in the original MOSES2.2 FAMOUS xfhcu (Williams et al., 2013) is close to that of fresh snow everywhere, with little spatial or temporal variation. This is attributable to the lack of significant grain-size evolution in the zero-layer snow model and to the chosen albedo being unrealistically high. In both respects, the HadRM3 configuration of UM4.5 is similar to FAMOUS xfhcu, and they contribute to the poor simulation of the GrIS SMB noted by Rae et al. (2012).

There are 4 areas where the snow and ice albedos in FAMOUS-ice have been modified:

1. The multi-layer snow scheme itself provides for a more effective coarsening of the snow pack grain-size compared to the old zero-layer snow model, and this improves the seasonal variation of albedo over snow surfaces globally. The increase in sensitivity of the grain-size may be because the thin surface layers are allowed to evolve independently with temperature, rather than having to change the full bulk of the snow pack.

2. For the grain-size-dependent calculation of snow albedo, parameters in the Marshall (1989) implementation of Wiscombe and Warren (1980) have been tuned to make the albedo lower for larger grains.

3. The albedo of the bare ice sub-surface has been tuned to lower values, and includes a dependency on air temperature as the surface rises towards $0°C$.

4. If there is a snow pack on an ice tile but the surface layers of the snow have a density approaching solid ice, then the bare ice albedo is used, rather than the albedo derived from the grain-size calculation, which is not appropriate for dense firn.

In JULES the albedo of seasonal snow, $a_{snow}$, is formulated in four discrete bands (for direct and diffuse components of both visible and near-infrared wavelengths) dependent on the snow grain-size, using the Marshall (1989) parameterisation. In FAMOUS-ice, for the visible wavelengths we have tuned the maximum snow albedo to 0.95 and increased the sensitivity to changes in surface grain-size, $\Delta a_{\mathrm{snow,visible}}$ so that our albedo has a grain-size dependency closer to the results of Roesch et al. (2002).

We found a range of values [0.006-0.008] $\mu\mathrm{m}^{-\frac{1}{2}}$ for $\Delta a_{\mathrm{snow,visible}}$ that all allow FAMOUS-ice to match the modern state of the GrIS but which produce different large-scale sensitivities of the ice sheet to changes in climate. This variation is explored further in Gregory et al. (2020). Simulations shown here use a mid-range value for $\Delta a_{\mathrm{snow,visible}}$ of 0.007 $\mu\mathrm{m}^{-\frac{1}{2}}$. The broadband albedo sensitivity to grain-size that results from our formulation is shown in figure 3.

The surface albedo on ice sheets, $a_{\mathrm{icesheet}}$, in FAMOUS-ice is formulated as

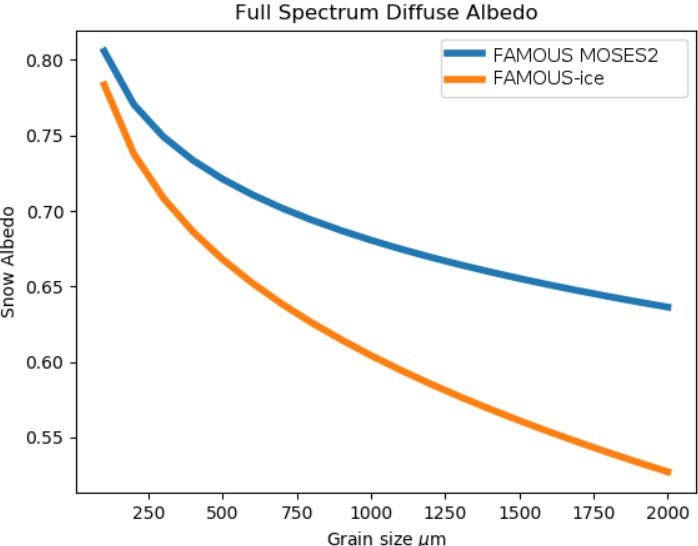

**Figure 3.** Variation of broadband albedo with surface snow grain-size for FAMOUS-ice (orange) and the previous version of FAMOUS (xfhcu, blue)

$$a_{\text{bare ice}} = \begin{cases} a_{\max} & T_{\text{surface}} \leq T_{\text{threshold}} \\ \max[(a_{\max} + (T_{\text{surface}} - T_{\text{threshold}}) \times \Delta a_{\text{ice}}), a_{\min}] & T_{\text{surface}} > T_{\text{threshold}} \end{cases}$$

$$f_{\text{snow}} = \begin{cases} 1 & \rho_{\text{surface}} < 600\text{kg/m}^3 \\ \max[(1 - \frac{\rho_{\text{surface}} - 600}{200}), 0] & \rho_{\text{surface}} > 600\text{kg/m}^3 \end{cases}$$

$$a_{\text{icesheet}} = f_{\text{snow}} \times a_{\text{snow}} + (1 - f_{\text{snow}}) \times a_{\text{bare ice}}$$

where $a_{\text{bare ice}}$ is the albedo of a bare ice surface, $a_{\max}$ is the maximum broadband albedo of a bare ice surface (set to 0.55), $a_{\min}$ is the minimum broadband albedo of a bare ice surface (set to 0.2), $\Delta a_{\text{ice}}$ is the sensitivity of bare ice albedo to surface temperatures once the surface is expected to melt (set to -0.35 K$^{-1}$), $T_{\text{surface}}$ is the surface temperature, $T_{\text{threshold}}$ is the threshold grid-box temperature above which melting is expected somewhere at the surface (set to 272 K), $\rho_{\text{surface}}$ is the density of the snow pack at the surface and $m_{\text{snow}}$ is the mass of the snow pack on the ice tile.

This formulation allows fresh snow on the ice sheet to have the same grain-dependent albedo parameterisation as for snow on non-ice sheet surfaces, but to transition to a lower albedos characteristic of bare ice surfaces when denser layers of the perennial snow pack are exposed at the surface. Bare ice albedo values have been tuned to match the range of ice albedos present in a MAR simulation of the modern GrIS. The temperature dependency of the ice albedo for temperatures above $T_{\text{threshold}}$ is a simple parametersation that mimics the presence of pooled melt-water or biological activity, which have been

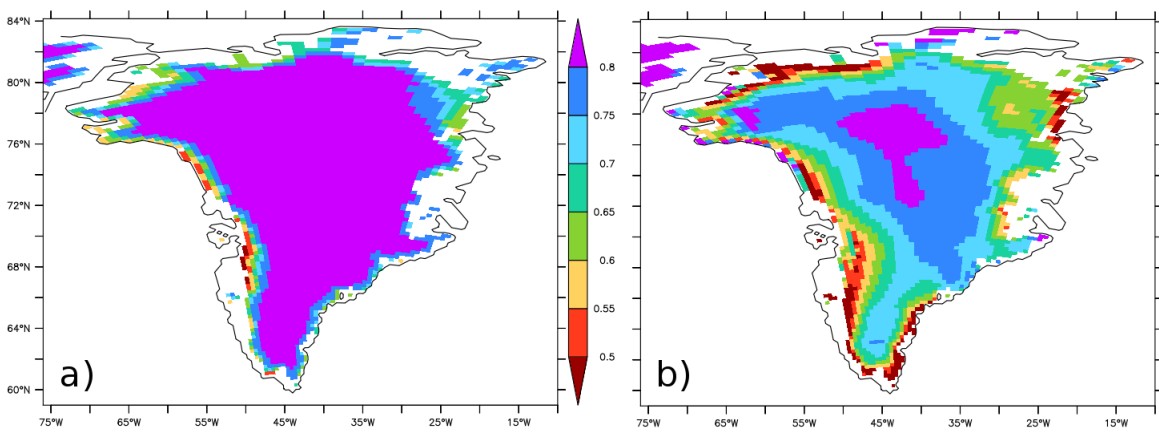

**Figure 4.** June-July-August average albedo for the surface of Greenland in a) MAR (minimum=0.48; maximum=0.83) (Fettweis et al., 2013) and b) FAMOUS-ice (minimum=0.23; maximum=0.78) forced by the MIROC5 climate (1980-1999). To visualise the distribution on sub-gridscale tiles, FAMOUS-ice results have been trilinearly mapped to the same topography as used in MAR.

shown to significantly lower surface albedo in certain regions of the ice sheet (Greuell, 2000; Williamson et al., 2020); these processes are not explicitly included in our surface scheme.

The resulting albedo for the GrIS in FAMOUS-ice does indeed produce lower values than in FAMOUS xfhcu at lower elevations and for melting surfaces. However, the FAMOUS-ice simulation now features a widespread low bias higher up on
the ice sheet during summer months when compared to MAR (figure 4). These low values apparently do not come from the grain-size albedo calculation, but from the transition to values more like those of bare ice which are used when surface snow density is high, following the melt of more recently fallen snow. The albedo is further lowered under the influence of relatively warm air across the ice sheet in summer, which triggers the parameterisation of pooling melt for these surfaces in regions that might not be appropriate in reality. Although there is some over-estimate of rainfall over the GrIS in FAMOUS (figure A9),
rain on ice tiles in FAMOUS-ice is directed straight to runoff, and can neither percolate into the snow nor pool on the surface, so this does not affect the albedo calculation. This widespread low albedo in FAMOUS-ice acts, in practice, to compensate for the reduced amount of incoming shortwave we simulate (figure A1), and results in the absorption of an equivalent amount of solar radiation as that seen in the RCMs (figure 2). This compensation could be seen as a direct outcome of our model tuning strategy, in which albedo parameters for snow and ice were varied as significant controls in achieving a stable GrIS shape and
an integrated SMB that matched RCM values. It should be noted that the configurations that use the end members of our range of tunings for $\Delta a_{\text{snow,visible}}$ also have a low-biassed albedo in their modern simulations; this low albedo bias seems a robust feature of FAMOUS-ice simulations that have a realistic ice sheet shape in the modern, but it does not necessarily imply a simple over- or under-estimate of the sensitivity of the albedo to future changes in climate.

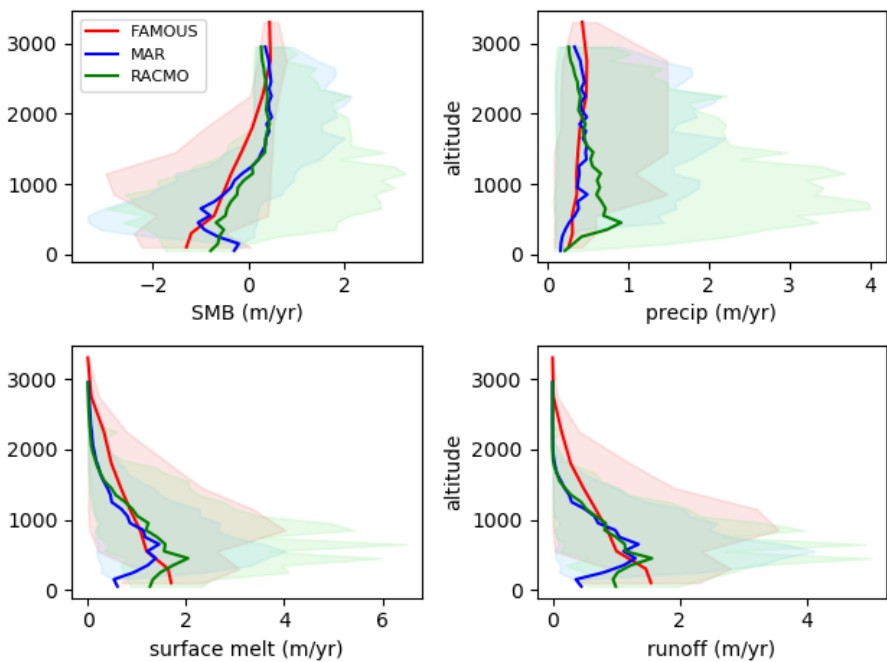

**Figure 5.** Profiles of surface mass balance, precipitation, surface melt and snow pack runoff taken from every surface modelled on the Greenland ice sheet binned by surface elevation. Red: FAMOUS-ice; blue: MAR (1980-1999) (Fettweis et al., 2013); green RACMO (1980-1999) (Noël et al., 2018). FAMOUS-ice and MAR are forced by the MIROC5 climate (1980-1999) (Watanabe et al., 2010), RACMO by ERA-interim (Dee et al., 2011). All quantities are annual averages. Shading represents the full range of values found in each time-averaged elevation bin.

## 7 Surface mass balance

### 7.1 Modern climate

The modifications we have described improve the simulated physical processes on Greenland in FAMOUS-ice, and consequently the overall simulation of SMB of the ice sheet. Further evaluation of the SMB simulation of FAMOUS-ice, along with

5 its sensitivity to some of the parameter choices in the snow scheme and to the simulated climate of the model, can be found in Gregory et al. (2020). To demonstrate the effect of our modifications, here we will compare SMB components from our illustrative FAMOUS-ice simulation with a central set of snow albedo parameters under a modern climate with the outputs from MAR (Fettweis et al., 2013) and RACMO (Noël et al., 2018), which are specialised regional climate models of Greenland. Sea surface temperature and sea ice surface boundary conditions used in FAMOUS-ice and MAR in these examples come from

10 the MIROC CMIP5 simulations (Watanabe et al., 2010), RACMO simulations were forced with ERA reanalysis climate (Dee et al., 2011) at their boundaries. It is to be expected that these RACMO simulation results will differ from the other models simply on the basis of the background climate forcing used.

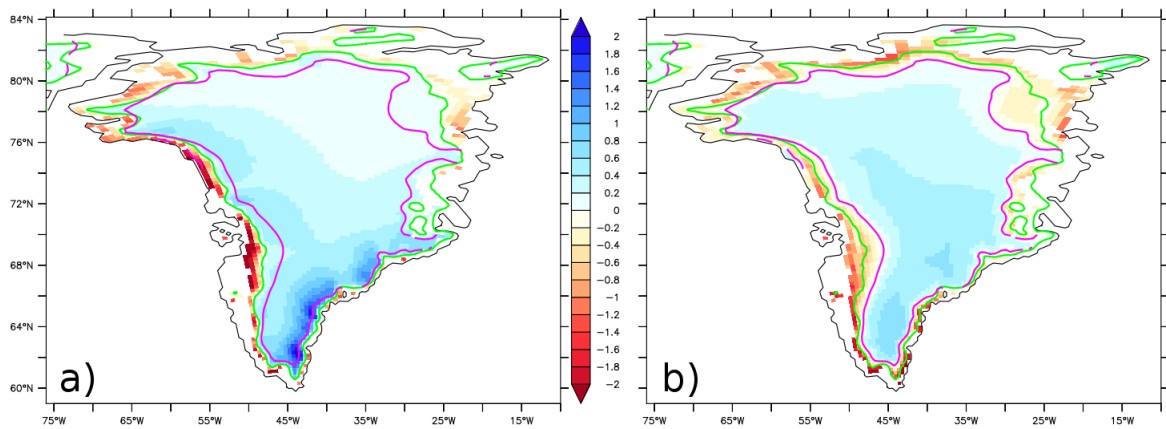

**Figure 6.** Annual average surface mass balance (m/yr LWE) for the surface of Greenland in a) MAR (1980-1999, minimum=-3.4; maximum=2.1) (Fettweis et al., 2013) and b) FAMOUS-ice (minimum=-4.3; maximum=0.83). To visualise the distribution on sub-gridscale tiles, FAMOUS-ice results have been trilinearly mapped to the same topography as used in MAR. The green contour shows the MAR equilibrium line, the pink is the FAMOUS-ice equilibrium line.

Binning SMB, precipitation and runoff into elevation classes shows that FAMOUS-ice simulates the variation of these quantities with height over the whole ice sheet well (figure 5) by comparison with MAR and RACMO. Seen in this average, FAMOUS-ice tends to produce an insufficiently positive SMB between 1000m and 2000m. The seasonal variation in these averages is of a similar range to the RCMs for positive mass balances, but overestimates the monthly variation in negative

SMB, especially at low elevations. SMB is primarily a balance between accumulation and liquid runoff from the snow pack, and it is apparent that the difference in the SMB profile between FAMOUS-ice and the RCMs can be attributed to the surface melt (and thus runoff) component. In both FAMOUS-ice and MAR there is relatively little variation in precipitation with height, and the magnitudes match well. In contrast, RACMO simulates more precipitation at low elevations and shows a steady reduction with height above around 500m, which largely accounts for the differences between the SMB-height profiles for

RACMO and MAR. This may be because of the higher native resolution of the RACMO simulation in this case resolving more orographic precipitation near the coast of Greenland.

The spatial pattern of the annual average SMB shows a clear correlation with the height of the ice sheet, as expected (figure 6). In general, the pattern diagnosed from FAMOUS-ice has smaller magnitudes than MAR in both accumulation and ablation zones. The relatively low spatial resolution of the FAMOUS atmosphere cannot simulate the same degree of intense

precipitation near the coasts, especially in the south east (figure A8). The equilibrium line altitude, demarking the transition between areas of net accumulation and ablation, is generally a little higher and further inland in FAMOUS-ice. This reflects the tendency of FAMOUS to both melt too far up the ice sheet and to distribute precipitation across a broad area rather than concentrate it on the slopes at the margins of the ice. In the south east and north west the FAMOUS equilibrium line is close to that of MAR but it differs significantly in other areas. It is further encouraging that FAMOUS-ice can simulate the magnitude

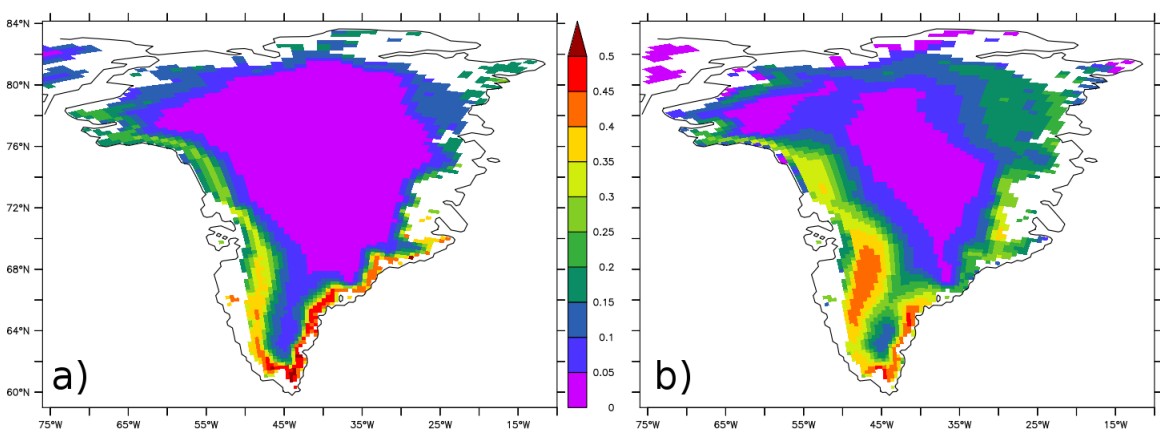

**Figure 7.** Annual average internal snow pack refreezing (m/yr LWE) for Greenland in a) MAR (1980-1999, maximum=0.59) (Fettweis et al., 2013) and b) FAMOUS-ice (maximum=0.56). To visualise the distribution on sub-gridscale tiles, FAMOUS-ice results have been trilinearly mapped to the same topography as used in MAR.

| Model | SMB | Snowfall | Surface Melt | Ablation | Refreeze |
|---|---|---|---|---|---|
| RACMO (ECMWF) | 406± 22 | 683 | 476 | 277 | 240 |
| MAR (ERA-Interim) | 388± 23 | 637 | 449 | 249 | 208 |
| MAR (MIROC5) | 437± 24 | 681 | 445 | 244 | 205 |
| FAMOUS-ice (MIROC5) | 342± 11 | 658 | 551 | 316 | 240 |
| change for the MIROC5 RCP45 climate 2080-2099 compared to 1980-1999 | | | | | |
| FAMOUS-ice | -163 | +18 | +295 | +181 | +113 |
| MAR | -240 | +18 | +331 | +258 | +84 |

**Table 1.** Ice sheet averaged simulated surface mass balance and relevant components for Greenland 1980-1999 in $\mathrm{Gt/yr}$LWE. MAR and RACMO data are from Fettweis et al. (2013) (using an earlier version of RACMO than that plotted in figures 2 and 5). ERA-interim and ECMWF climate forcing for MAR and RACMO comes from Dee et al. (2011), MIROC5 climate forcing comes from Watanabe et al. (2010). SMB is the total change in snow pack mass over a year. Ablation is the difference between snowfall and SMB, so includes all processes that remove mass from the snow pack such as sublimation. Since FAMOUS-ice does not route rainfall through the snow, refreeze has been defined under the assumption that all of the rain falling on the snow has run off. The ± uncertainty shown for SMB is the standard error of the time-mean, estimated by assuming annual values to be independent.

and the basic spatial distribution of internal refreeze of melt within the snow pack (figure 7). Anomalies in this field reflect the biases seen in the SMB field, with refreezing too far inland in both the south and the north of the GrIS, reflecting the fact that FAMOUS simulates summer melt over too much of the ice sheet in the modern, rather than confining melt to the coasts. Distributions of other components of SMB are provided in supplementary figures A8-A11.

Integrated over the ice sheet as a whole, SMB and its components simulated for the modern climate are in reasonable agreement with RCMs (table 1), although there is a high bias in melt that produces a lower overall SMB than in the RCMs. FAMOUS–ice reports smaller annual variability in SMB than the RCMs as the climatological SSTs used in FAMOUS–ice exclude interannual variability.

## 7.2 Future climate

It is important to understand the sensitivity of model processes to likely forcings as well as their equilibrium behaviour. Although these can be described for large-scale coupled climate-ice models, their realism can be difficult to evaluate since many of these feedbacks have not been observed in detail as they play out. Gregory et al. (2020) show that the sensitivity of FAMOUS-ice GrIS-integrated SMB under a range of future climate scenarios is a reasonable match to the cubic relationship fitted by Fettweis et al. (2013) to results from MAR. Here we document in more detail how some of the processes in FAMOUS-ice change under the moderate RCP4.5 climate forcing scenario (van Vuuren et al., 2011). The example FAMOUS-ice simulation is forced by a climatology of SST and sea-ice fields from the MIROC5 RCP4.5 simulation (Watanabe et al., 2010), averaged over the period 2080-2099, with atmospheric greenhouse gas concentrations from the same period. We compare with data averaged over the same period from a transient simulation of MAR, forced by boundary conditions from the same MIROC5 run (Fettweis et al., 2013).

This FAMOUS-ice configuration simulates a smaller reduction in SMB than MAR for the RCP4.5 warming scenario (table 1). The two models see similar increases in net snowfall, but FAMOUS-ice produces less additional melt at the surface, and a greater proportion of that melt refreezes in the snow pack rather than running off. The recent GrSMBMIP exercise (Fettweis et al., 2020) concluded that runoff biases in simulations of Greenland SMB models for the recent climate would likely persist and be exacerbated under future warming, but this does not seem to be the case for FAMOUS-ice where melt in the recent climate is greater than in MAR, but the change in FAMOUS-ice melt for the future is less than in MAR.

The pattern of the GrIS SMB change in response to the warmer climate is quite different in MAR than FAMOUS-ice. FAMOUS-ice SMB becomes more negative almost everywhere, with larger decreases in the south west and north east regions of the ice sheet (figure 8). In contrast, MAR simulates more intense negative SMB in a narrow band confined to the edges of the ice sheet. As suggested by table 1, these different SMB responses are more clearly attributable to different patterns of melt and refreeze than to changes in snowfall (supplementary figures A13 - A15). The two models also simulate very different changes in downwelling shortwave at the surface of the GrIS in response to the RCP4.5 climate, suggesting that local cloud responses between the two models are not similar (figure A16). GrIS surface air temperature changes differ between the two models as well (figure A12), with FAMOUS-ice seeing much more surface warming at higher elevations. This indicates a reduction in the near-surface atmospheric lapse rate as FAMOUS warms, and is also linked with an increase in net absorption of shortwave radiation, with a reduction in albedo in the north and east of the ice sheet convolving with the change in downwelling shortwave.

The modern-day simulation of this FAMOUS-ice configuration has a lower albedo at the margin of the GrIS than MAR (figure 4), so some of the margin is already near the lowest albedos that ice can realistically have. Under a moderate climate change scenario like RCP4.5 it is thus possible for MAR to simulate larger albedo reductions of the lowest-lying parts of the ice

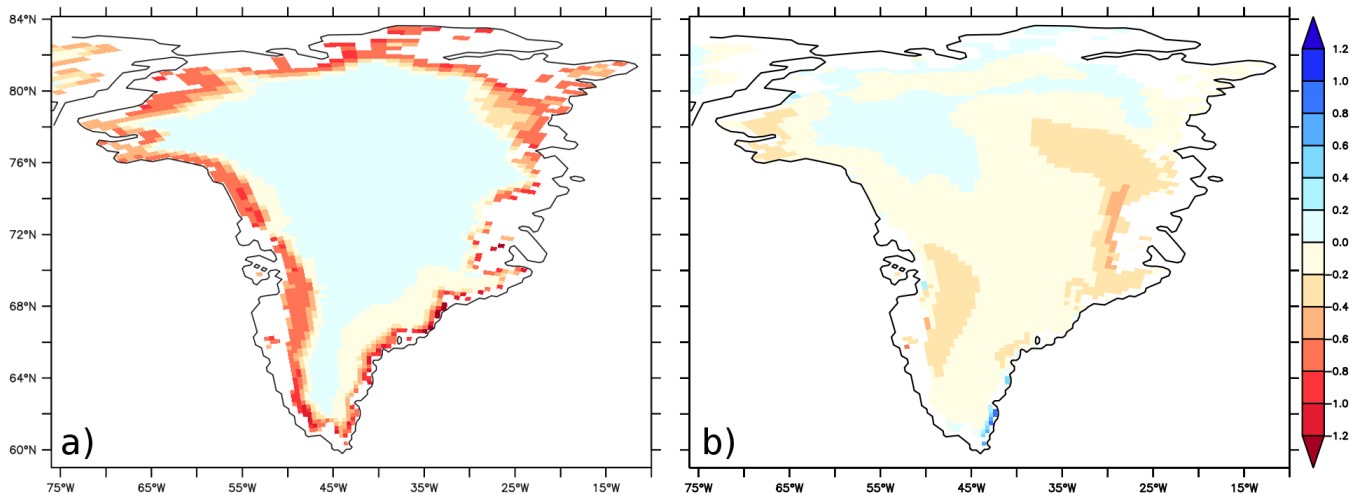

**Figure 8.** Change in annual average surface mass balance (m/yr LWE) for the surface of Greenland between the MIROC5 climate 2080-2099 under RCP4.5 and the 1980-1999 climate shown in figure 6. a) MAR (minimum change=1.48; maximum=0.08), b) FAMOUS-ice (minimum change=-0.73; maximum=0.92). To visualise the distribution on sub-gridscale tiles, FAMOUS-ice results have been trilinearly mapped to the same topography as used in MAR.

margin, contributing to differences in the pattern of SMB simulated in these two models. However, this difference in potential albedo response only affects the very edge of the ice and does not necessarily dominate the SMB response over longer periods of time (when prolonged large negative SMB near the ice margin could remove those areas from the ice sheet and coupled flow dynamics become important) or under a stronger climate forcing affecting larger areas of ice.

5      Gregory et al. (2020) find that reductions in GrIS surface shortwave in response to the changing shape of the ice sheet plays an important role in a negative feedback between future GrIS mass loss and SMB. The pattern and magnitude of future regional anomalies in climate are uncertain, with changes in cloud and thus radiation being one of the most uncertain aspects. When considering the long-term future of the ice evolution, the area integrated mass balance is more important than its gradients in a coupled model, where ice dynamics will respond to determine the shape of the ice sheet. The changes we simulate in the 10   magnitude of GrIS surface radiation under the RCP4.5 future climate scenario are in line with the range simulated by CMIP5 climate models, although these MAR and FAMOUS simulations sit at opposites extremes of that range.

## 8   Coupling with Glimmer

Our primary motivation for the model development work we have described is to enable FAMOUS-ice to carry out centennial coupled climate-ice sheet simulations. As in previous studies with FAMOUS (Gregory et al., 2012; Roberts et al., 2014), we 15   use the Glimmer ISM (Rutt et al., 2009). More recent versions of Glimmer (Glimmer-CISM, Lipscomb et al., 2019), are part of

the ice sheet model within the CESM (Danabasoglu et al., 2020). For computational efficiency we use Glimmer in its original shallow-ice formulation without sliding and a fixed internal temperature profile, although the climate model coupling itself does not preclude the use of more sophisticated ice sheet or solid earth physics. In this section we concentrate on technical aspects of coupling FAMOUS-ice and Glimmer directly. The results of a suite of simulations exploring the future evolution of the climate and the GrIS in this coupled system are described in Gregory et al. (2020).

For coupling, we use our modified JULES multi-layer snow scheme to represent the surface and upper levels of the ice sheet, down to a depth beyond which seasonal climate variations are not important. Thus the top of the ice sheet is considered to be within the domain of FAMOUS, and the dynamic ice underneath is handled by Glimmer. Transfer of information between FAMOUS and Glimmer consists of annual average fields, passed once a year between the bottom of the FAMOUS snow pack at the interface with Glimmer. This proceeds in a series of steps, as follows.

## 8.1 Diagnosis of SMB in FAMOUS-ice

SMB at each elevation is diagnosed as the change in snow pack mass in FAMOUS-ice over the course of a year. The absolute mass of the snow pack is irrelevant in this context, as long as it is deep enough to not become exhausted over the year.

At each annual coupling step, once the SMB has been calculated for Glimmer, the FAMOUS snow pack mass is reset to its initial value. This procedure in effect passes mass from FAMOUS to Glimmer or vice versa. An accumulation of snow pack mass in FAMOUS indicates a positive mass tendency for the ice sheet, a loss of snow pack mass (i.e. ablation) a negative tendency. The situation where there is not enough mass available in Glimmer to refill the FAMOUS snow pack (i.e the ice sheet has retreated from a location) will be addressed in section 8.4. In this way, mass is transferred between the models, and the tiles representing the ice sheet in FAMOUS-ice will always have enough snow present to diagnose an SMB term for Glimmer during the next year of simulation. The amount chosen for the ice sheet snow pack is 100 m of liquid water equivalent (LWE). In the coupled model, we initialise the snow pack on ice sheet tiles as ice-density snow. During the model spin-up, fresh snow builds up on the surface in the accumulation zone and mass is removed by the coupling at the base of the snow pack, while in the ablation zone ice melts at the surface and the coupling adds mass from Glimmer at the base of the snow pack.

## 8.2 Interpolation of SMB to the Glimmer grid

The SMB field on the elevation classes received by Glimmer is transferred onto the Glimmer topography using the same trilinear interpolation as in Vizcaíno et al. (2013). The full SMB field for each elevation class is first bilinearly interpolated onto to horizontal locations of the Glimmer grid. Each point on the Glimmer grid then interpolates vertically between the grids of the elevation classes above and below its topographical height. If the Glimmer topography is below the mid-height of the lowest FAMOUS elevation class, or above the mid-height of the highest elevation class, then no vertical interpolation can be done and the value from the relevant elevation class is used unaltered. The same interpolation procedure is applied to the temperature field at the bottom of the FAMOUS snow packs, to provide a surface temperature boundary condition for Glimmer.

Linear interpolation is not generally conservative, and the interpolated output can be globally scaled to ensure that the total mass given to Glimmer matches the mass change in FAMOUS-ice, although this has not been done in the illustrative

simulations here. Non-conservation may also arise from the area of the ice sheet being differently represented in each model e.g. due to mismatches of coastline, in which case imposing a numerical scaling on the SMB seen by the ice sheet model to make it conform to the SMB in FAMOUS-ice may not provide the best estimate of the surface forcing that the climate would really produce for an ice sheet of that precise shape. A local adjustment to conserve the mass change of the Glimmer points

located in each FAMOUS-ice grid-box was also tested, but was found to heavily imprint the outline of the 5°x7.5° FAMOUS-ice grid on the 20km Glimmer SMB field. We thus do not use this correction in FAMOUS-ice, but more acceptable results would be expected if such a technique were used with a less-coarse grid climate model as a source. In practice, in the example simulations shown in section 7, the non-conservation due to regridding between the models was less than 2% of integrated GrIS SMB taken over a 10 year mean.

Using these boundary conditions from FAMOUS-ice, Glimmer simulates the evolution of the ice sheet state. For synchronous coupling with the climate, Glimmer simulates a year of ice flow before transferring information back to FAMOUS. However, under the assumptions that the change in ice sheet geometry over $N$ years, where $N$ is a small number, is too small to have a significant effect on the climate and SMB calculated by FAMOUS-ice, and that the global climate is either constant or changes negligibly within $N$ years, it is also possible to use an asynchronous coupling scheme where the ice is allowed to evolve for $N$

years under the same climate boundary conditions. In our experience, running $N =10$ ice sheet years for every year of climate simulated does not  significantly affect the evolution of the coupled climate-ice system, but running N=100:1 does result in differences compared to a synchronously coupled run.

## 8.3    Update of FAMOUS following Glimmer

After timestepping the ice sheet for the required number of years, Glimmer passes several fields to FAMOUS: the mean

orographic height for each FAMOUS-ice grid-box; the fractions of ice and non-ice covered areas in each elevation class to define the tile fractions in FAMOUS-ice, and some fields describing the sub-gridscale distribution of the orography within each grid-box for use by parameterisations of atmospheric gravity-wave and boundary layer drag. All these are calculated from the updated topography on the Glimmer grid.

Glimmer not only changes the height of the ice sheet, but can also produce ice in formerly unglaciated areas or make areas

ice-free if the SMB provided is sufficiently negative. All of these effects are represented in FAMOUS-ice through changes in the area fractions of the sub-gridscale tiles, which in turn affect the grid-box mean surface properties. For structural reasons within FAMOUS, it is not possible to change the land-sea mask of the climate model as it runs. Glimmer may thus only move the ice within the boundaries of FAMOUS-ice's pre-defined Greenland land area, and our coupled system is unable to simulate the growth of floating shelves, displacement of ocean by advancing grounded ice or changes in sea-level. Any ice that Glimmer

moves beyond the coastline defined by FAMOUS-ice is calved to the ocean.

## 8.4    Creating or destroying glaciated grid-boxes

An unglaciated grid-box in Glimmer may be occupied by ice in one of two ways. Glimmer may move ice into it, or it may accumulate a sufficient depth of meteoric snow to pass into the regime where we would want Glimmer to be able to treat it

dynamically. To allow this latter process, information about the snow mass on unglaciated tiles in FAMOUS is also passed to Glimmer and interpolated in the same trilinear fashion as the SMB. Where the resultant field has a snow depth greater than a given threshold on a grid-box that is not already occupied by ice in Glimmer then a new ice point is initialised. To conserve mass, the unglaciated tile in FAMOUS-ice that sourced this snow has an equivalent mass subtracted from it.

Conversely, as the ice in Glimmer thins then we need a method for deglaciating grid-boxes. When the ice becomes so thin it is no longer dynamically active and potentially unable to fully resupply the FAMOUS snow pack at the start of a coupling period we remove all ice from that grid-box in Glimmer and increase the area fraction of the corresponding unglaciated tile in FAMOUS-ice accordingly. The mass of ice removed from Glimmer is also added to the snow pack in the unglaciated tile in FAMOUS-ice. The threshold for creation or removal of ice from Glimmer is set at 50m liquid water equivalent in the example
simulation used here.

Since snow in FAMOUS is held as a mass-per-area term, a change to the area fraction of a tile will result in a change in the absolute mass of snow on that tile, regardless of any additional adjustments that may also be required by converting FAMOUS snow to Glimmer ice (or vice versa) when Glimmer grid-boxes are (de)glaciated. Full conservation of the multi-layered, multi-prognostic snow pack properties as the ice sheet area waxes or wanes is a complex operation. There are many science questions
that do not require a model to be able to initiate dynamically active ice in new areas purely from meteoric snow, nor which require water mass to be conserved exactly across the system components. In such cases we allow ice to spread to new areas only by Glimmer ice sheet dynamics, and diagnose the degree of non-conservation implied by not adjusting the snow packs to account for changes in area fraction.

### 8.5 A simulation of the modern Greenland Ice Sheet in FAMOUS-ice

It has been shown that the SMB calculation of FAMOUS-ice for the GrIS in a modern day climate compares reasonably well to other models. Once coupled to a ice sheet model however, biases in the simulated SMB may result in an unacceptable simulation of the dynamics and geometry of the ice.

In a run of 5000 years of ice sheet evolution (500 years of climate simulation and an asynchronous time factor of N=10) of FAMOUS-ice coupled to Glimmer forced by the same modern greenhouse gases and sea surface conditions described in
section 7, GrIS volume and area grow to reach a steady state about 10% larger than observed (figures 9, 10). The resolution and shallow-ice numerics of Glimmer mean that individual outlet streams and Greenland fjords are not well simulated, and the requirement that ice only calves when it reaches the coast of the climate model means that the ice sheet expands toward the coast. As it does so it slumps in the centre and builds up higher shoulders at the margins to allow enough ice to calve to maintain balance with the climate model SMB, which is not intense enough right at the edge of the ice sheet, especially in the
south (figure 9, red lines, figures 10a, A17a). Too much ablation in the far north results in thinning at this edge of the GrIS, whilst the overestimate of accumulation in the north east sector of the ice sheet produces thickening. When calving is instead enforced at the current ice-edge, making up for the lack of explicit fjord calving at this resolution of Glimmer, the ice sheet does not spread in this way and the volume is also more realistic (figure 9, black lines, figures 10b, A17b) but the pattern of thickening and thinning tendencies is similar.

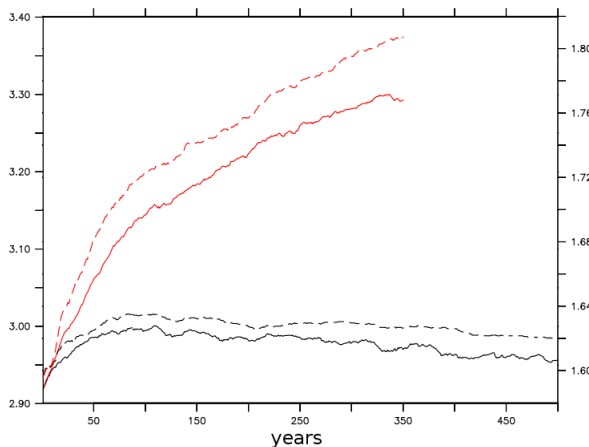

**Figure 9.** Spinup of the Greenland ice sheet in the coupled FAMOUS ice. Ice sheet volume (x10$^6$km$^3$ LWE, solid lines, left axis) and area (x10$^6$km$^2$, dashed lines, right axis). Black: calving imposed near the observed ice edge; red: ice sheet allowed to calve at the coast. Asynchronous coupling was used: the time-axis shows climate model years, the ice sheet evolves 10 years for every 1 year of climate.

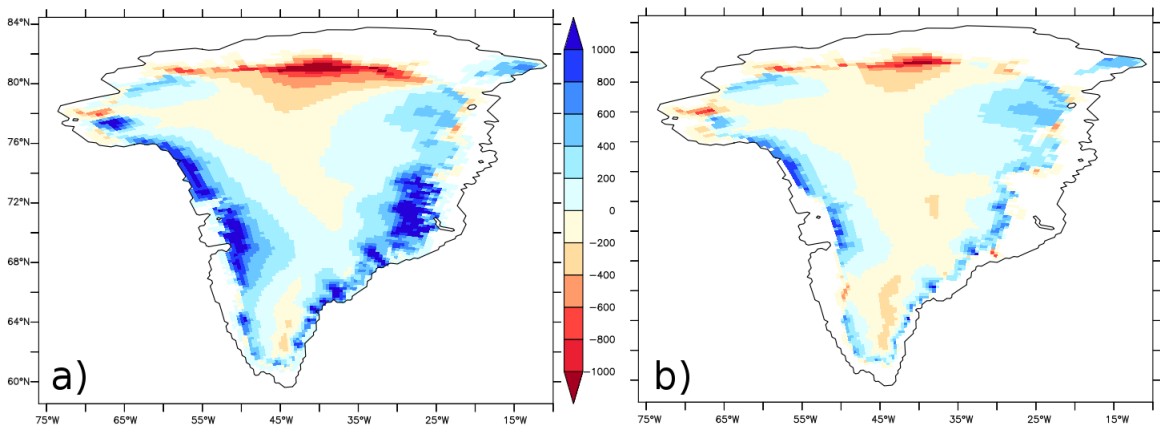

**Figure 10.** Change in Glimmer ice thickness (m) at the end of the spinup periods shown in figure 9. Calving is imposed either (a) at the marine margin as resolved on the Glimmer grid or (b) near the initial ice edge.

More details of the simulation of the GrIS in coupled FAMOUS-ice and its sensitivity to climate change can be found in Gregory et al. (2020).

## 9   Summary and Discussion

We have described modifications to the FAMOUS climate model that enable it to calculate a realistic surface mass balance for
ice sheets, and its coupling to an model of dynamic ice sheet flow. Together, these developments allow coupled simulations of the multi-millennial evolution of climate and grounded ice sheets to be carried out with more fidelity to the physical processes in reality.

There are a number of biases in the detail of the downscaled surface climate for the GrIS in FAMOUS-ice, judged by comparison with much higher resolution RCMs. Although this paper is not intended as a climate evaluation exercise, the
question arises as to whether these biases are rooted more in the large-scale climate of FAMOUS or arise from our approach to downscaling. Drawing such a clear line between these two options is rather artificial, and shortcomings in both are likely to feedback on each other in a model such as ours, but it does highlight some issues worthy of discussion. A very similar approach to downscaling for ice sheet surface mass balance - even similar down to the level of the model code - is taken in UKESM1 (Sellar et al. (2019), Smith et al. (2021), submitted ms.) at a much higher resolution. GrIS surface fields in UKESM1 are
generally much more similar to those in RCMs than the FAMOUS fields shown here, suggesting that the downscaled process parameterisations in the FAMOUS-ice surface scheme are fundamentally capable of doing better, but are being held back by other factors in FAMOUS-ice, either the background climate simulation or the horizontal resolution they are required to work at.

As an example of a background climate factor, precipitation is not downscaled to elevation tiles in FAMOUS-ice, and is
a pure output of the rather coarse atmosphere model. Global GCM atmospheres are well-known for producing widespread, persistent drizzle (Stephens et al., 2010) even in modern, higher resolution models, and the smoothed representation of the GrIS orography and relatively high levels of numerical diffusivity inherent to the low resolution FAMOUS grid exacerbate this tendency. Being able to robustly downscale precipitation is a clear avenue for improvement in FAMOUS-ice although we have not found a way to do this well via a simple correlation with tile height, let alone whilst retaining water and energy conservation
with respect to the atmosphere model. The significant bias in downwelling shortwave radiation over the GrIS (figure A1) is also a direct product of the FAMOUS atmosphere, and cannot be attributed to failings in the surface modelling. The low albedo biases (figures 2, 4) could be seen as shortcomings in the surface scheme, but since these partly result from a tuning approach to compensate for the bias in downwelling shortwave radiation, even the albedo bias cannot simply be attributed to a failure in our surface downscaling approach. These factors all point to biases in the background climate of FAMOUS playing a dominant
role in determining how well the surface downscaling can perform.

If the downscaling itself, or at least its implementation on the FAMOUS grid, were thought to be the root of our biases, then changing the resolution of the downscaled surface might be seen as an avenue for improving model performance. It is hard to change the horizontal resolution of the downscaled surface grid independently from the atmosphere above, but increasing the

number of elevation classes could be done quite easily. However, on the evidence we have it is unlikely that additional classes, even with potential retuning of the surface parameters, would make a major difference to our results or the biases present in the surface fields. Early version of FAMOUS-ice used 25 classes, and no major degradation in scientific performance was seen when 10 levels were adopted. Ultimately, to produce a "good" set of surface fields from a biased background climate

would require significant, deliberately unphysical modelling interventions, such as using flux adjustments. That would not be an appropriate strategy for FAMOUS-ice, intended to simulate climates very different from the modern observed climate, and a balance must be struck between accepting model biases that are present and some tuning that makes allowances for them. Recognising the biases in climate in our example here, work to comprehensively tune and characterise the behaviour of FAMOUS-ice across large ensembles of parameter-perturbed simulations is ongoing.

FAMOUS-ice has been used for studies of the future evolution of the GrIS, including an analysis of the ice sheet's stability and the presence of tipping points (Gregory et al., 2020). This same framework will be used for paleoclimate studies, where the co-evolution of and feedbacks between the climate and ice sheets are essential to understanding the glacial cycles of the last millennia. A necessary future development is the ability to simulate sea-level rise and the spread of ice onto coastal shelves, which is currently not possible as the land-sea mask in FAMOUS is fixed.

The configuration of FAMOUS-ice described in this paper is limited to simulating ice sheets that are predominantly grounded and not subject to direct influence from contact with the ocean. The shallow ice approximation used here in Glimmer is also not suited to simulating streaming ice. Techniques for coupling models of floating ice shelves and marine-terminating glaciers to the ocean is a subject of current research (e.g. Asay-Davis et al., 2016), and it is not yet clear how this should best be done in any modelling system, let alone at coarse spatial resolution and over centennial or millennial timescales. Therefore, whilst

FAMOUS-ice can justifiably be applied to Greenland and the Laurentide ice sheet of the Quaternary glacial cycles, it cannot simply be used to simulate Antarctica or other paleo ice sheets which are predominantly affected by interactions with the ocean. It would be possible to use a parameterisation of the influence of ocean temperature on neighbouring ice sheets, such as in Favier et al. (2019), and this will be pursued in future work. Elevation tiles are also not as effective for downscaling a coarse-resolution climate simulation on Antarctica, where surface conditions are less tightly dependent on altitude than on Greenland,

and more determined by small-scale dynamic processes such as katabatic winds, blowing snow and synoptic variability. In this case accumulation needs to be redistributed both vertically between tiles and horizontally between grid-boxes.

One of the key features of FAMOUS as a model is its ability to run multi-millennial climate simulations using modest amounts of computational resource compared to modern AOGCMs. The need to investigate more feedbacks in the Earth system over these timescales was in fact one of the main motivations for this present work. The substantial technical developments

presented in this paper do increase the computational cost of FAMOUS, but not so much as to make such simulations impractical. Previous versions of FAMOUS (e.g. xfxwb (Smith et al., 2014), very similar to configurations used in most published studies with FAMOUS) simulate around 250 years of climate per day on 6 processors of the Reading University Academic Computing Cluster. The additional complexities in FAMOUS-ice-Glimmer (MOSES2.2, the multi-layer snow scheme, climate downscaling and Glimmer) reduce throughput to around 165 years of climate, or 1650 years of ice sheet evolution per day for

asynchronous coupling with N=10. These figures are intended only to be illustrative of the relative cost of the different model

versions and throughput could be increased further by using more cores, although the coarse resolution means that FAMOUS does not scale well beyond 16 processors. IO costs are also significant in FAMOUS as all output must be routed through and processed on single core, and some of the additional cost of FAMOUS-ice is purely down to the additional size of new fields dimensioned across multiple snow layers and elevation tiles; these costs cannot be reduced by using more processors.

The code for the coupled system we have developed for FAMOUS-ice may be used with some other models. HadCM3 (Gordon et al., 2000) and the HadRM3 and PRECIS regional models (Jones et al., 2004) rely on the same UM4.5 code-base as FAMOUS, and it would be possible to directly transfer the adaptations to the tiles and the multi-layer snow scheme to these models. HadCM3 has a better climate simulation than FAMOUS and is still used extensively in the UK climate research community, although it is more computationally expensive. On the ice side, the FAMOUS-ice coupling to Glimmer could

alternatively accommodate the BISICLES ice sheet model (Cornford et al., 2013). BISICLES is a variable resolution, adaptive mesh ISM whose numerics allow it to simulate ice streams, floating ice shelves and grounding line retreat in a computationally affordable manner. Using BISICLES would remove the restrictions inherent in Glimmer's shallow-ice approximation.

Scientific questions around understanding and projecting sea-level rise under future climate change are receiving increased attention from the community, with international efforts such as MISOMIP (Asay-Davis et al., 2016) and ISMIP6 (Nowicki

et al., 2016) recognising the fundamental importance of coupled ocean, climate and ice processes. The new UKESM1 (Sellar et al., 2019) model includes a configuration capable of interactively simulating the GrIS using elevation tiles in the climate model in a manner very similar to that described here. Although the climate simulation of UKESM1 contains considerably more physical detail than FAMOUS-ice, UKESM1's computational cost makes it impossible to conduct the multi-millennial climate simulations that are needed to resolve many questions of the co-evolution of climate and ice sheets. Simpler models

such as FAMOUS are still needed for many of the scientific questions we urgently need to answer about how the Earth system functions.

**Code availability**

FAMOUS is a configuration of the Met Office Unified Model, whose code is owned by the Met Office and protected under UK Crown Copyright. Glimmer-CISM code is owned jointly by the Universities of Bristol, Edinburgh, Swansea, Montana

and the Los Alamos National Laboratory, and is copyrighted under the GNU Lesser General Public License. The underlying source code for version 4.5 of the Portable UM (PUM) on which FAMOUS is built can be viewed at http://cms.ncas.ac.uk/ code_browsers/UM4.5/UMbrowser. Modifications to this source code are commonly packaged as patch files known as mods. The mods required to configure the PUM as FAMOUS-ice can be downloaded from https://puma.nerc.ac.uk/svn/UniCiCles_svn/ UniCiCles/tags/FAMOUS-ice_SEG, along with a fork of version 1.9 of Glimmer-CISM it can couple with. Namelists and

configuration details that specify the FAMOUS-ice simulation illustrated in this work are available via job xotzt in the central UMUI service provided by NCAS CMS at puma.nerc.ac.uk. Additional support for obtaining and running the PUM and FAMOUS-ice simulations in practice is available via the NCAS CMS helpdesk at http://cms.ncas.ac.uk/wiki/CmsHelpdesk or via the authors.

## Data availability

Simulation output from the FAMOUS-ice run used in the illustrative plots can be downloaded from http://gws-access.jasmin.ac.uk/public/ncas_climate/rssmith/sgfjb

## Author contributions

5  RS and JG designed the coupling schemes, which were coded in FAMOUS and Glimmer by RS and SG. SG improved the model and conducted the simulations. All authors contributed to the simulation analysis. The manuscript was written by RS.

## Acknowledgements

We would like to acknowledge Richard Essery, Jamie Rae, Jeff Ridley and Andy Wiltshire for authoring and providing additional UM code in helpful formats. Output from the MAR and RACMO regional models was kindly made available by Xavier

10  Fettweis and Brice Noel. We would also like to acknowledge two anonymous reviewers, whose comments helped us improve this paper. The work was supported by NERC grants NE/I011099/1 *Modelling ice-sheets, climate and sea-level during the last glacial cycle* and NE/P014976/1 *Thresholds for the future of the Greenland ice-sheet* and the National Centre for Atmospheric Science.

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

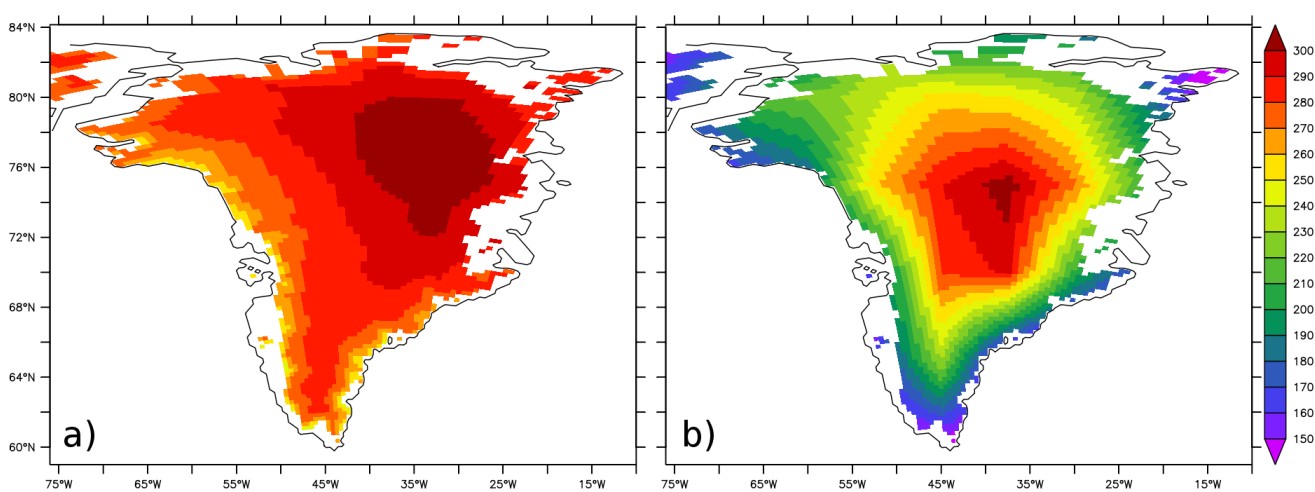

**Figure A1.** June-July-August average downwelling shortwave radiation at the surface ($W/m^2$) for the surface of Greenland in a) MAR (1980-1999, minimum=242; maximum=306) (Fettweis et al., 2013) and b) FAMOUS-ice (minimum=102; maximum=307). To visualise the distribution on sub-gridscale tiles, FAMOUS-ice results have been trilinearly mapped to the same topography as used in MAR.

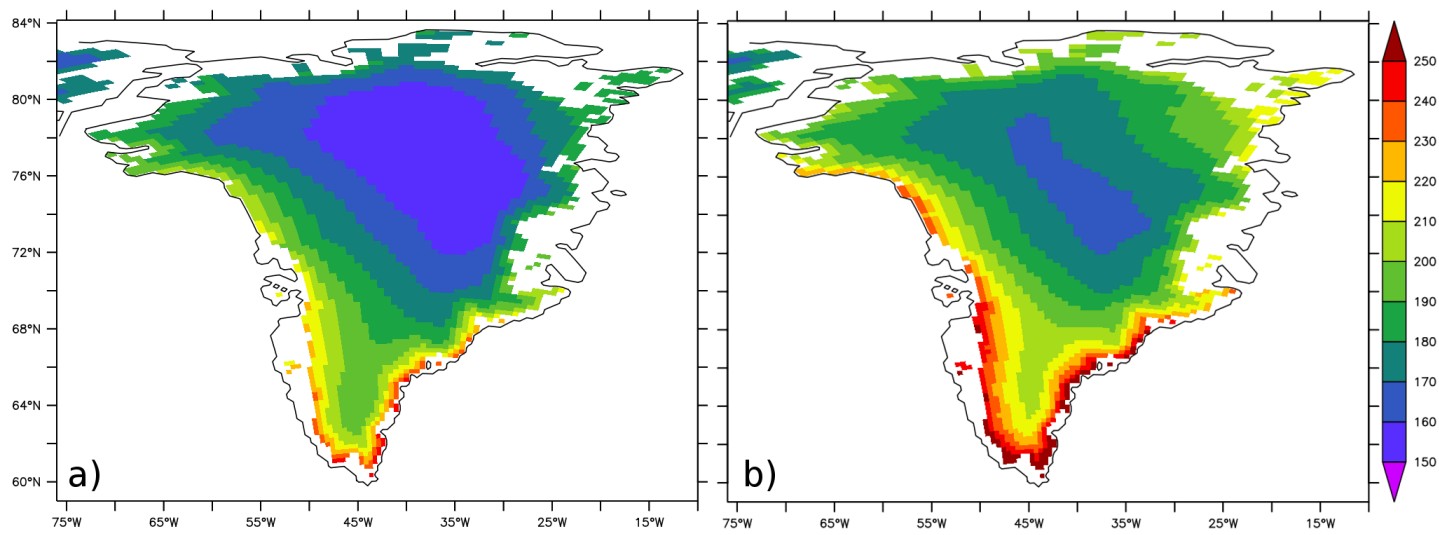

**Figure A2.** Annual average downwelling surface longwave radiation (W/m$^2$) for the surface of Greenland in a) MAR (1980-1999, minimum=152; maximum=248) (Fettweis et al., 2013) and b) FAMOUS-ice (minimum=155; maximum=285). To visualise the distribution on sub-gridscale tiles, FAMOUS-ice results have been trilinearly mapped to the same topography as used in MAR.

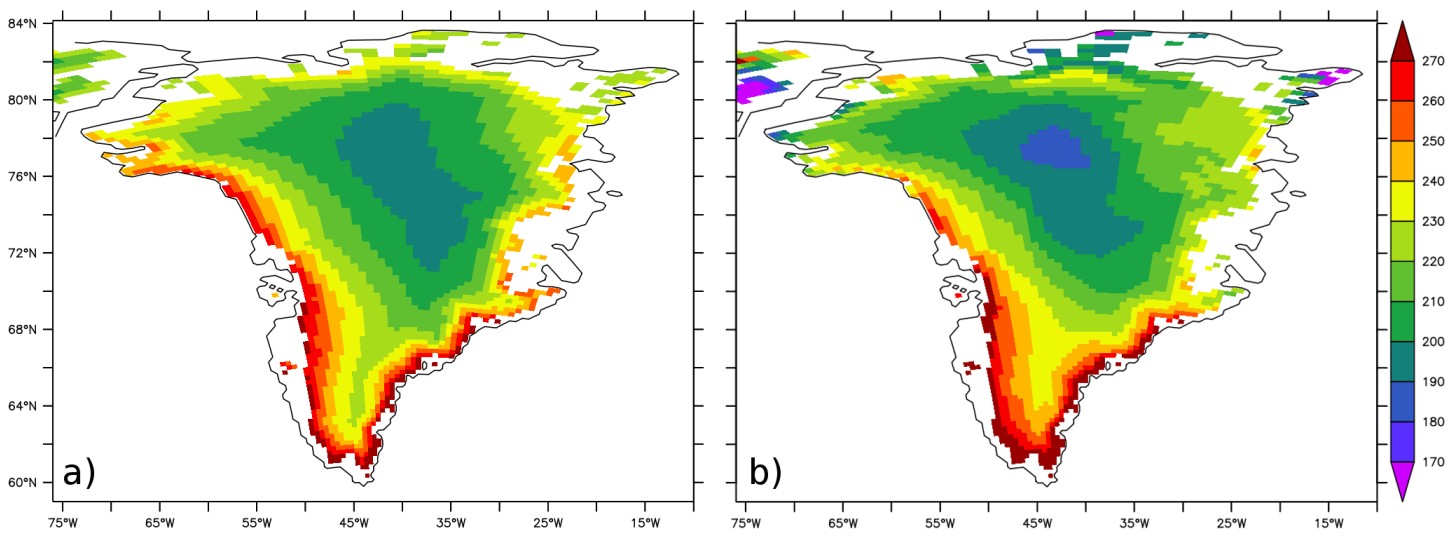

**Figure A3.** Annual average upwelling surface longwave radiation (W/m$^2$) for the surface of Greenland in a) MAR (1980-1999, minimum=196; maximum=289) (Fettweis et al., 2013) and b) FAMOUS-ice (minimum=184; maximum=302, on the body of the ice sheet). To visualise the distribution on sub-gridscale tiles, FAMOUS-ice results have been trilinearly mapped to the same topography as used in MAR.

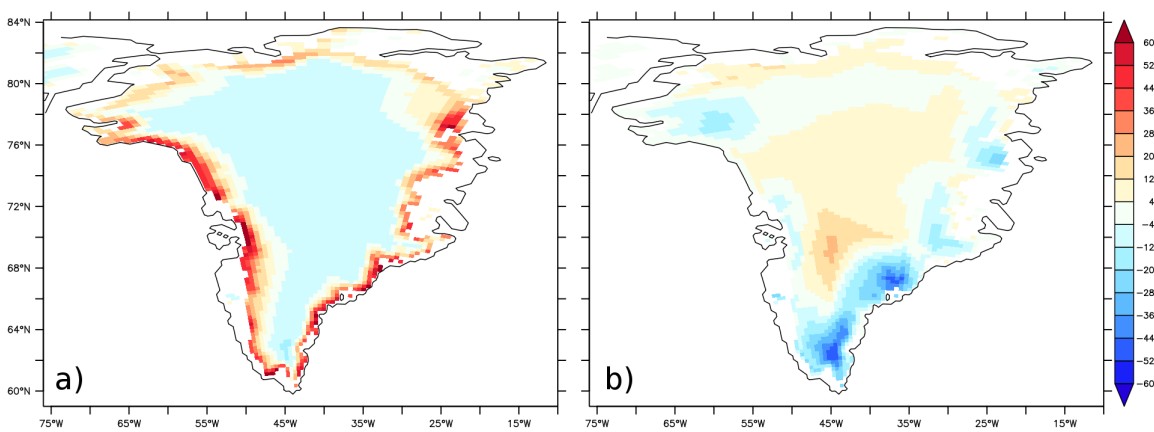

**Figure A4.** Annual average sublimation (mm/yr LWE) for the surface of Greenland in a) MAR (1980-1999, minimum=-13.11; maximum=76.41) (Fettweis et al., 2013) and b) FAMOUS-ice (minimum=-115.4; maximum=44.9). To visualise the distribution on sub-gridscale tiles, FAMOUS-ice results have been trilinearly mapped to the same topography as used in MAR.

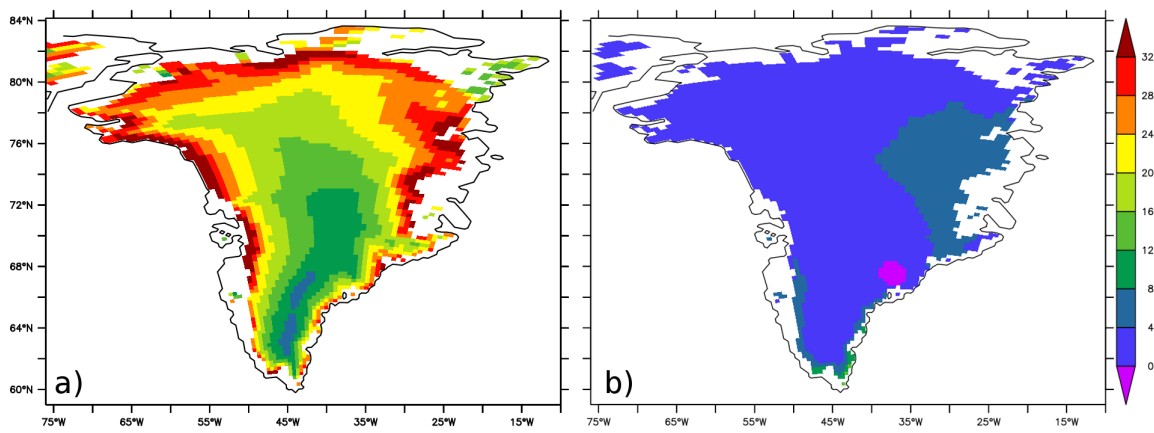

**Figure A5.** Annual average sensible heat flux $(W/m^2)$ for the surface of Greenland in a) MAR (1980-1999, minimum=5.12; maximum=41.83) (Fettweis et al., 2013) and b) FAMOUS-ice (minimum=-5.5; maximum=15.7). To visualise the distribution on sub-gridscale tiles, FAMOUS-ice results have been trilinearly mapped to the same topography as used in MAR.

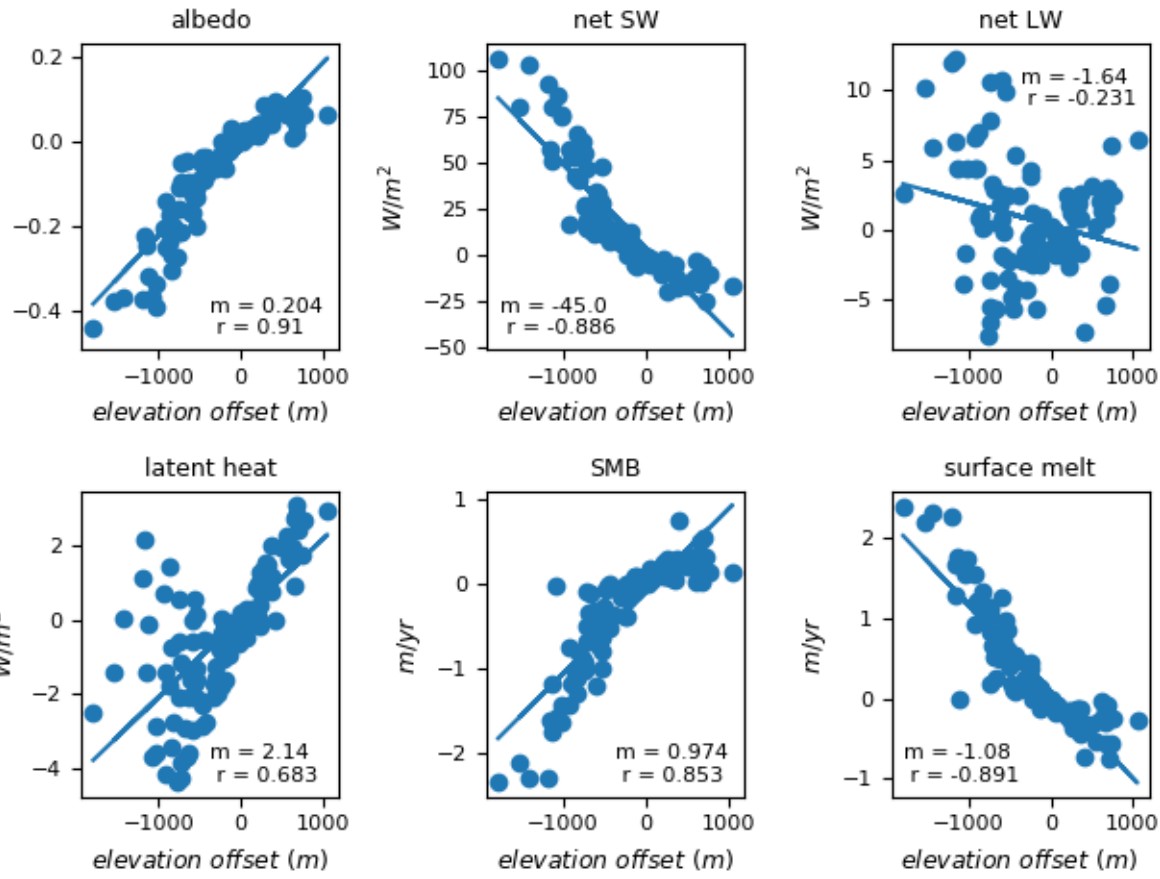

**Figure A6.** This figure shows how quantities relevant to surface mass balance modelled on the elevation tiles depend on their height difference from the grid-box mean altitude. A linear relationship is inferred by regression, whose slope ($m$, units/km) and correlation coefficient ($r$) are given on each panel. All panels show JJA-average relationships, except for SMB and surface melt which are annual average. These can be compared with figures 1 and 3 of (Sellevold et al., 2019), which show equivalent data for the elevation classes in CESM1 and higher resolution representations in RACMO.

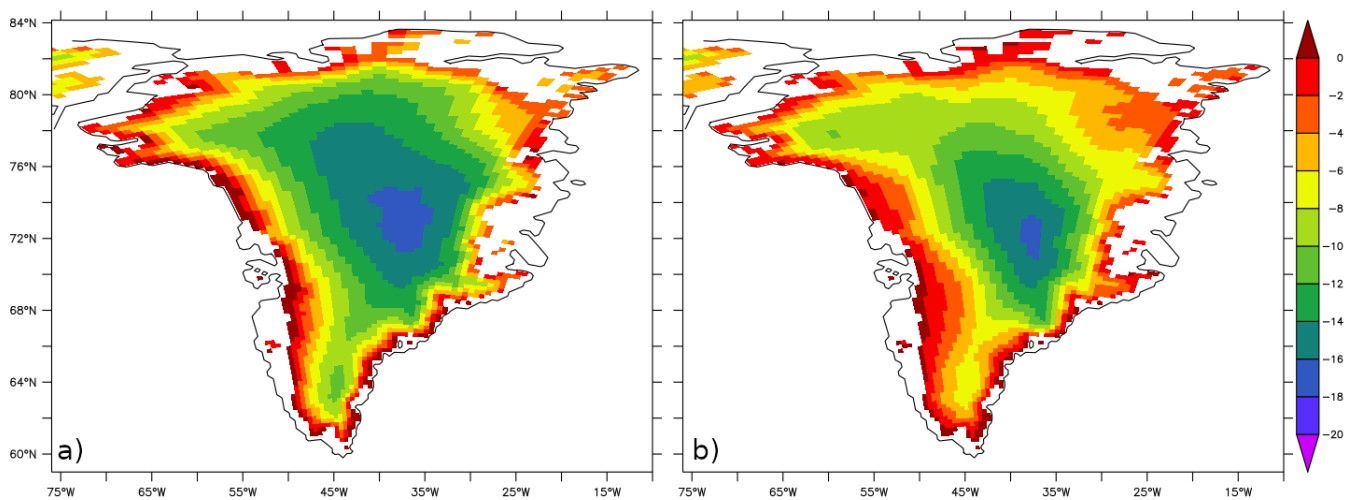

**Figure A7.** June-July-August average air temperature (°C) for the surface of Greenland in a) MAR (3m temperature, 1980-1999, minimum=-16.55; maximum=3.578) (Fettweis et al., 2013) and b) FAMOUS-ice (1.5m temperature, minimum=-16.54; maximum=1.913). To visualise the distribution on sub-gridscale tiles, FAMOUS-ice results have been trilinearly mapped to the same topography as used in MAR.

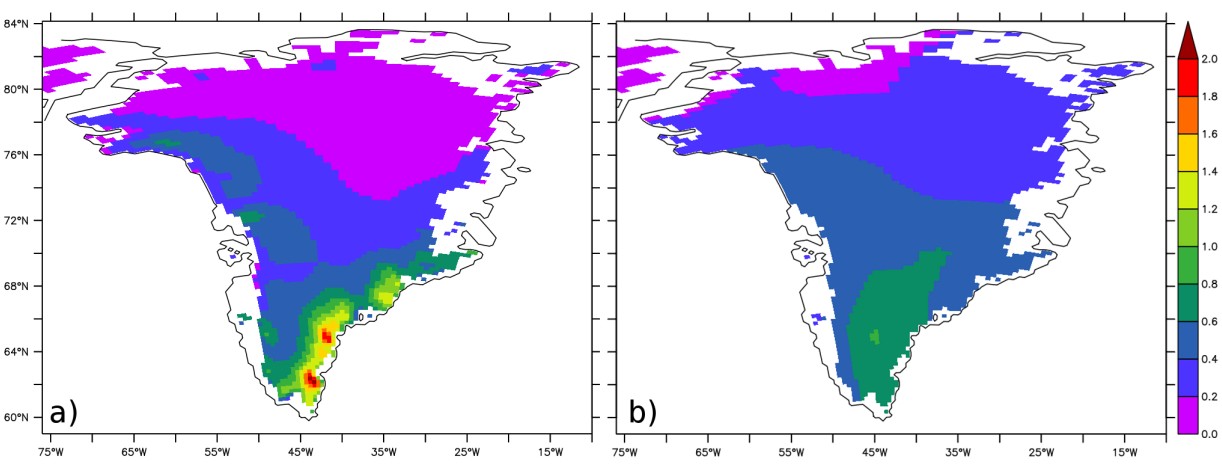

**Figure A8.** Annual average snowfall (m/yr LWE) for the surface of Greenland in a) MAR (1980-1999, minimum=0.08; maximum=2.12) (Fettweis et al., 2013) and b) FAMOUS-ice (minimum=0.0; maximum=0.83). To visualise the distribution on sub-gridscale tiles, FAMOUS-ice results have been trilinearly mapped to the same topography as used in MAR.

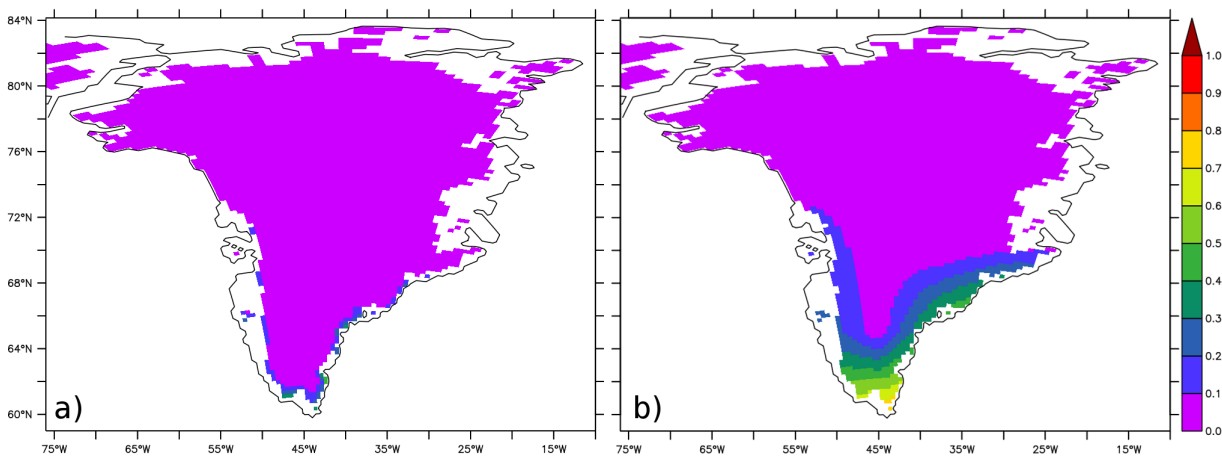

**Figure A9.** Annual average rainfall (m/yr LWE) for the surface of Greenland in a) MAR (1980-1999, minimum=0.0; maximum=0.45) (Fettweis et al., 2013) and b) FAMOUS-ice (minimum=0.0; maximum=0.77). To visualise the distribution on sub-gridscale tiles, FAMOUS-ice results have been trilinearly mapped to the same topography as used in MAR.

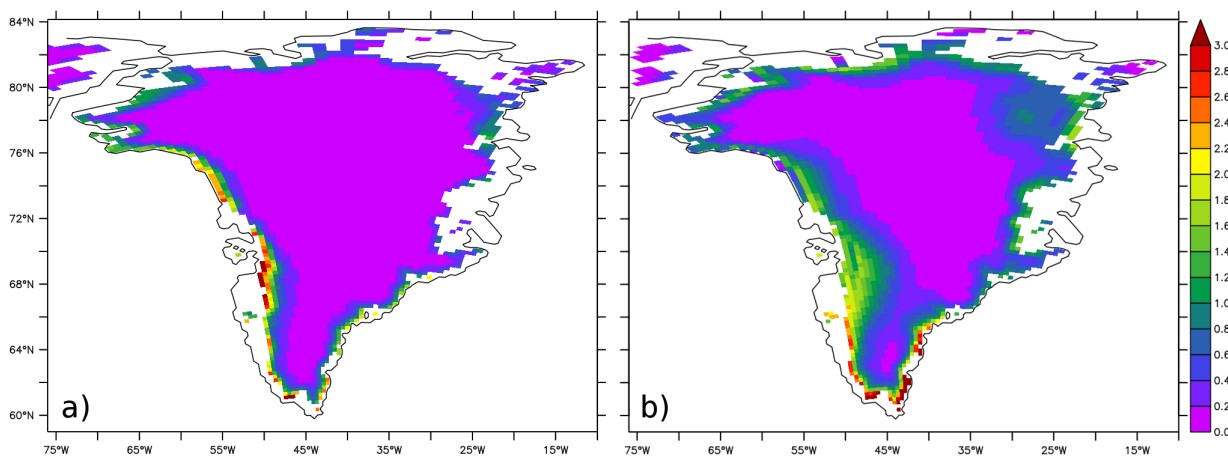

**Figure A10.** Annual average surface melt (m/yr LWE) for the surface of Greenland in a) MAR (1980-1999, minimum=0.0; maximum=4.06) (Fettweis et al., 2013) and b) FAMOUS-ice (minimum=0.0; maximum=5.45). To visualise the distribution on sub-gridscale tiles, FAMOUS-ice results have been trilinearly mapped to the same topography as used in MAR.

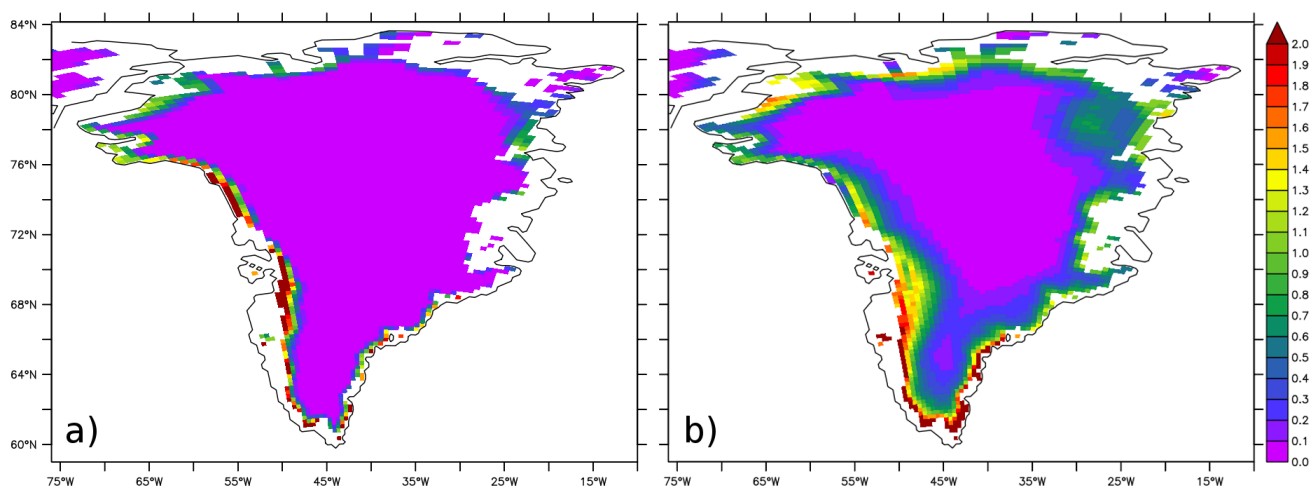

**Figure A11.** Annual average runoff (m/yr LWE) for the surface of Greenland in a) MAR (1980-1999, minimum=0.0; maximum=4.09) (Fettweis et al., 2013) and b) FAMOUS-ice (minimum=0; maximum=5.84). To visualise the distribution on sub-gridscale tiles, FAMOUS-ice results have been trilinearly mapped to the same topography as used in MAR.

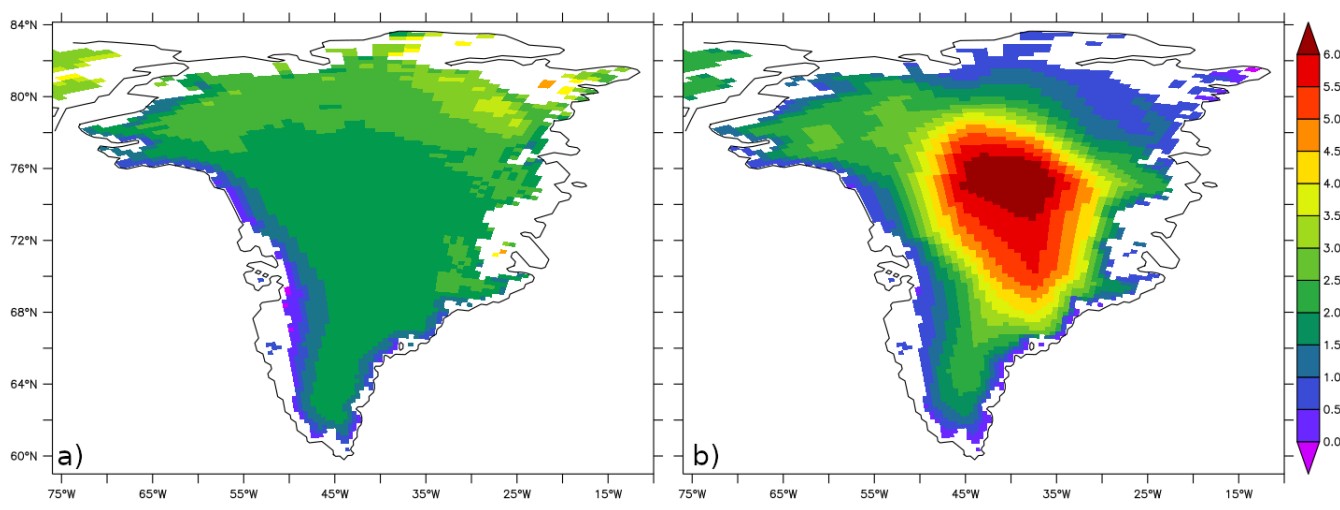

**Figure A12.** Change in June-July-August air temperature ($^{\circ}$C) for the surface of Greenland between the MIROC5 climate 2080-2099 under RCP4.5 and the 1980-1999 climate in figure A7. a) MAR (3m temperature, minimum change=0.44; maximum=4.83), b) FAMOUS-ice (1.5m temperature, minimum change=-0.02; maximum=6.838). To visualise the distribution on sub-gridscale tiles, FAMOUS-ice results have been trilinearly mapped to the same topography as used in MAR.

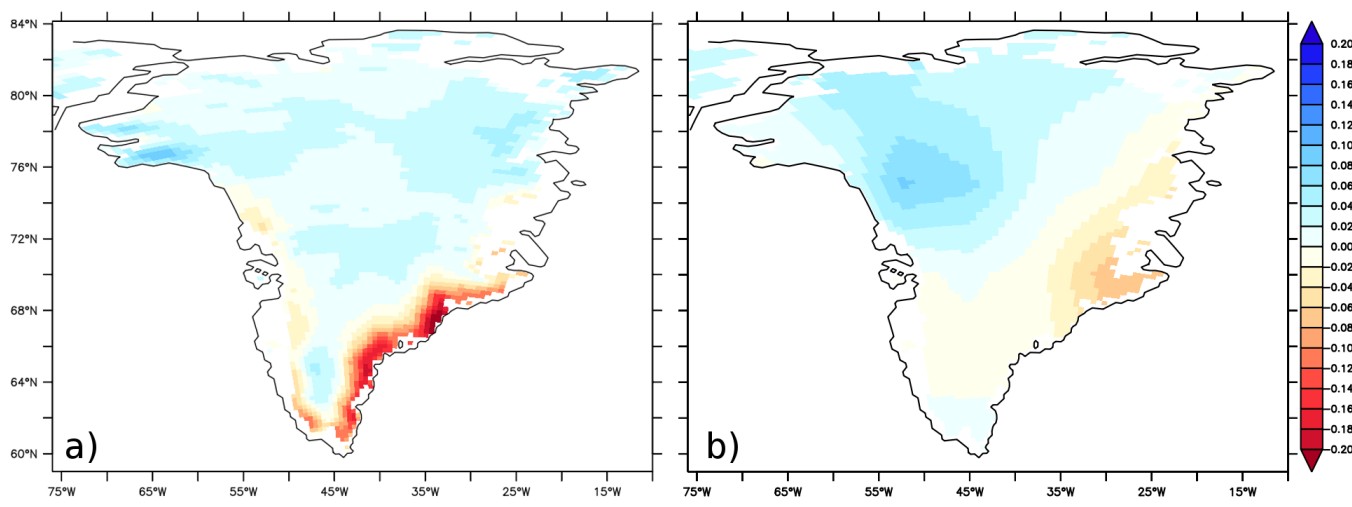

**Figure A13.** Change in annual average snowfall (m/yr LWE) for the surface of Greenland between the MIROC5 climate 2080-2099 under RCP4.5 and the 1980-1999 climate in figure A8. a) MAR (minimum change=-0.26; maximum=0.10), b) FAMOUS-ice (minimum change=-0.07; maximum=0.08). To visualise the distribution on sub-gridscale tiles, FAMOUS-ice results have been trilinearly mapped to the same topography as used in MAR.

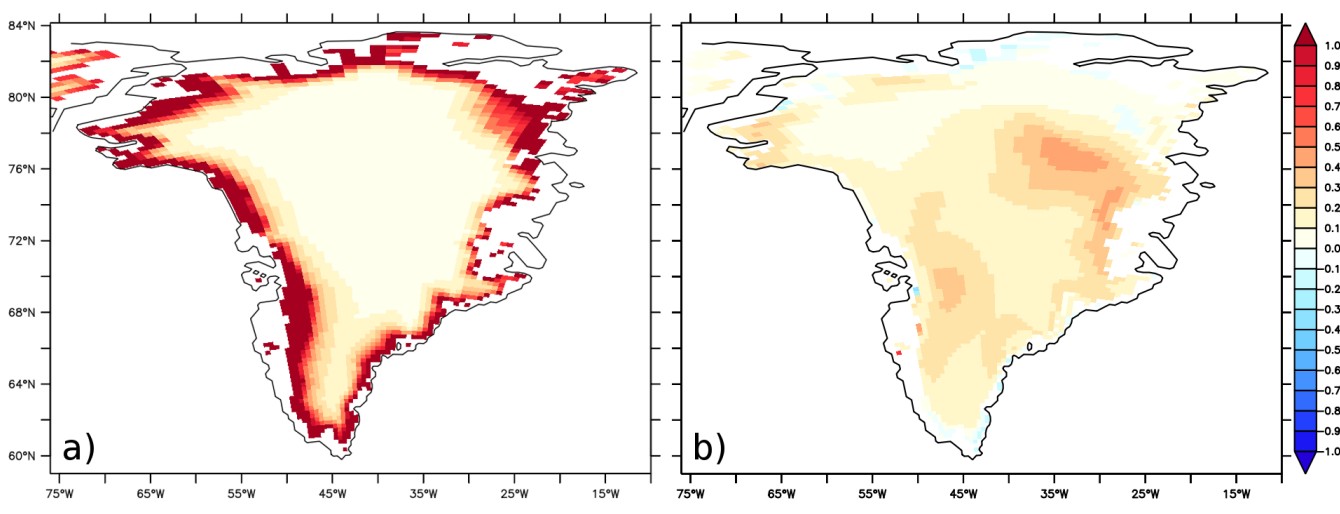

**Figure A14.** Change in annual average surface melt (m/yr LWE) for the surface of Greenland between the MIROC5 climate 2080-2099 under RCP4.5 and the 1980-1999 climate in figure A10. a) MAR (minimum change=0.0; maximum=1.14), b) FAMOUS-ice (minimum change=-0.35; maximum=0.75). To visualise the distribution on sub-gridscale tiles, FAMOUS-ice results have been trilinearly mapped to the same topography as used in MAR.

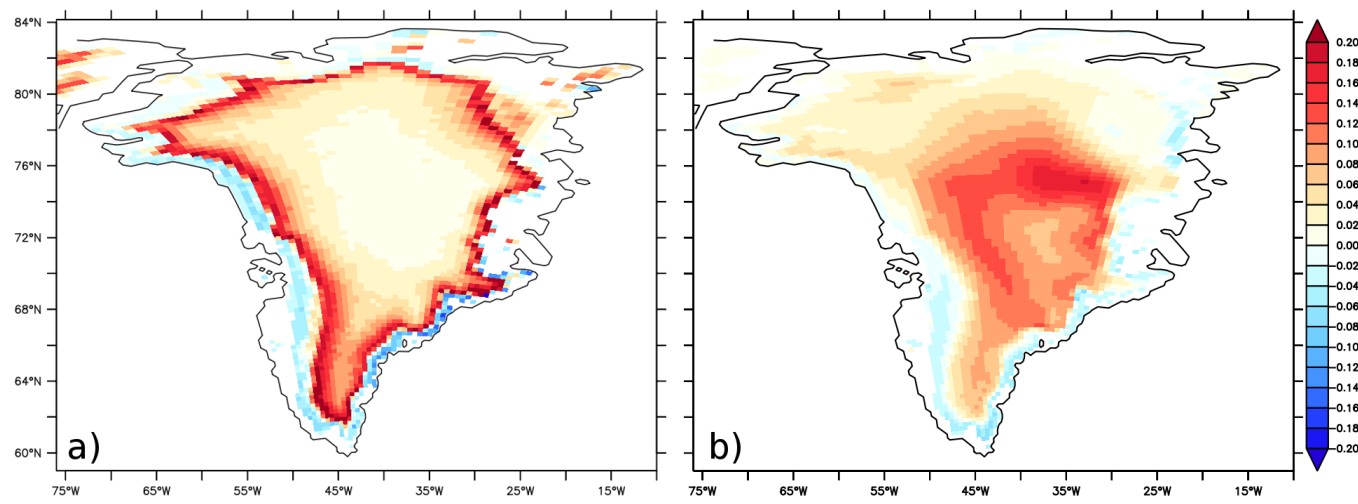

**Figure A15.** Change in annual average snow pack refreezing (m/yr LWE) for the surface of Greenland between the MIROC5 climate 2080-2099 under RCP4.5 and the 1980-1999 climate in figure 7. a) MAR (minimum change=-0.32; maximum=0.27), b) FAMOUS-ice (minimum change=-0.07; maximum=0.18). To visualise the distribution on sub-gridscale tiles, FAMOUS-ice results have been trilinearly mapped to the same topography as used in MAR.

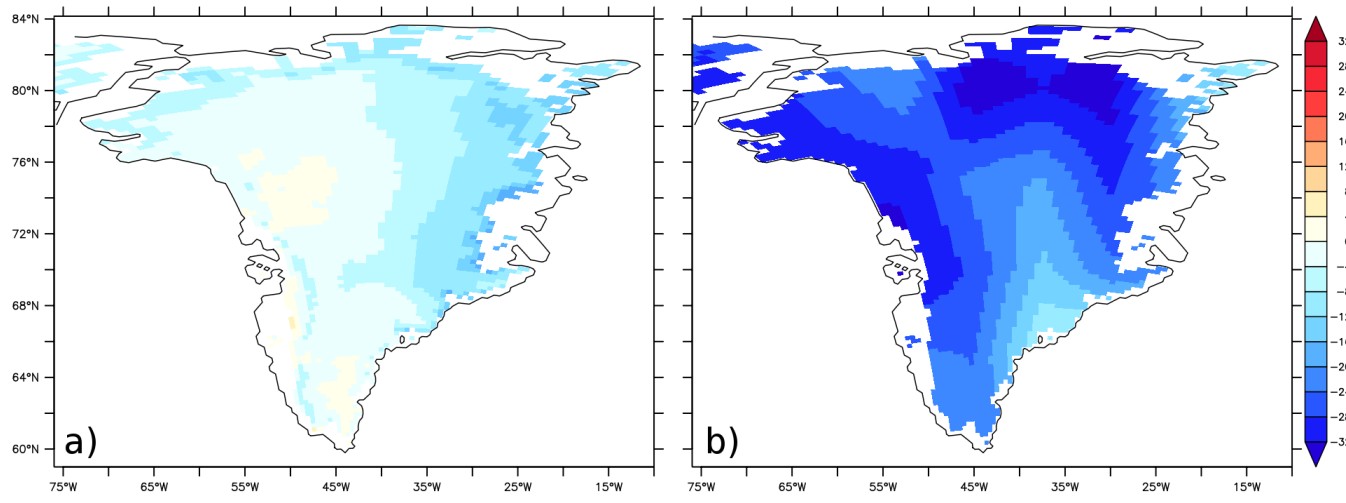

**Figure A16.** Change in June-July-August downwelling shortwave ($W/m^2$) for the surface of Greenland between the MIROC5 climate 2080-2099 under RCP4.5 and the 1980-1999 climate in figure A1. a) MAR (minimum change=-19.73; maximum=5.70), b) FAMOUS-ice (minimum change=-35.32; maximum=-8.30). To visualise the distribution on sub-gridscale tiles, FAMOUS-ice results have been trilinearly mapped to the same topography as used in MAR.

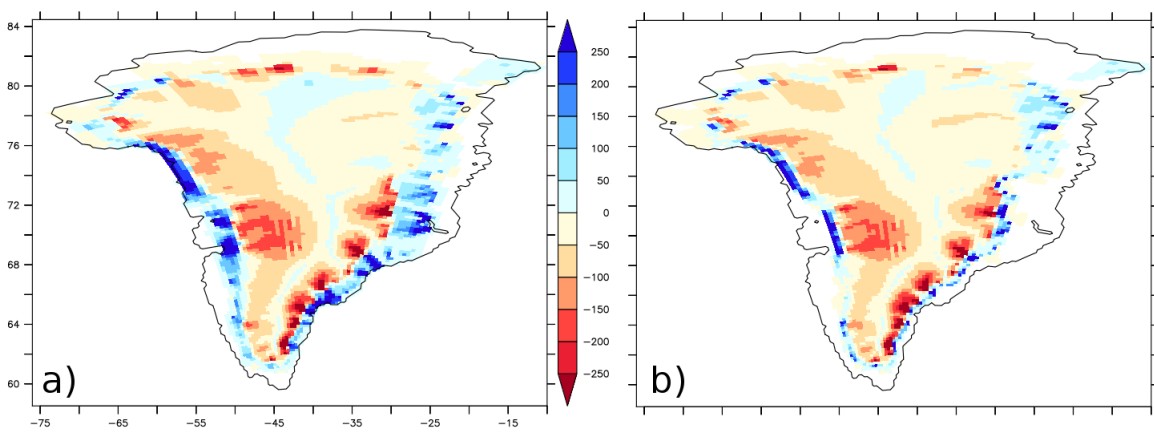

**Figure A17.** Change in Glimmer ice velocity (m/yr) at the end of the spinup periods shown in figure 9. Calving is imposed either (a) at the marine margin as resolved on the Glimmer grid or (b) near the initial ice edge.