# Peer review of "FAMOUS version xotzt (FAMOUS-ice): a GCM capable of energyand water- conserving coupling to an ice sheet model"

_Geoscientific Model Development, 2020_

## Referee Comment (RC1) · Anonymous Referee #1 · 1 Sep 2020

**1   General comments**

This manuscript presents the Earth System Model (ESM) FAMOUS-ice, which is designed for coupled ice sheet-climate simulations on multi-millennial time scales. Branching off from FAMOUS-xfhcu, this model configuraion has been extended by a multi-layer representation of the snow-pack. Furthermore the model is now capable to consider glaciated surfaces on multiple subgridscale-tiles of different elevation for each model grid-box.

The manuscript consists of a description of these modifications to FAMOUS and evaluates simulated components of the surface mass balance (SMB) and the resulting long-term coupled response of the Greeenland Ice Sheet (GrIS) based on a simulation forced by sea surface and sea ice conditions from a CMIP5 simulation.

The inclusion of realistic interactions between ice sheets and the climate system in ESMs is timely and an important step towards a more comprehensive view on long-term climate change. The hard-wired implementation of a snow pack model which is resolving sub-gridscale topography is a promising approach, which is capable to resolve essential small scale processes in the context of long-term simulations. FAMOUS is a paricularly fast model and is one of the few general circulation models which can cover glacial-interglacial timescales. As such the presented model is a particularly valuable contribution but the presented results also indicate considerable shortcomings in the representation of the spatial distribution of the surface mass balance, which, in my opinion, should be addressed in more depth. Seeing these problems with spatial characteristics it is even more important to analyse the model's SMB response to temporal climate variations — this is missing in the manuscript.

Furthermore the manuscript could benefit from introducing the general concept of the snow-pack scheme, which is here only described where it differs from a referenced published version.

Before publication I recommend major revisons (see comments below).

**2   Major comments**

The presented analysis demonstrates that FAMOUS-ice in combination with an ice sheet model is capable to simulate a reasonably realistic GrIS under present day climate conditions. The surface mass balance, however, appears to be extended to too high elevations and albedo seems to be overestimated in the ice sheet's interior. Given that other ESMs of similar design and resolution exhibit a qualitatively better skill (e.g.

Kapsch et al., 2020; Fettweis et al., 2020), the analysis of the model should answer the question, whether these shortcomings are the result of a biased climate or of a misrepresentation of specific processes at the ice sheet's surface.

Fig. 2 indicates that the downward shortwave radiation is strongly biased towards lower values while downward longwave radiation does not exhibit any clear bias. As argued by the authors, this may indicate an overestimated cloud cover, which should be substantiated by decomposing long wave radiation into atmospheric emissivity and near surface temperature. A potentially biased cloud cover should be reflected in atmospheric emissivity. Near surface temperature influences both, downward longwave radiation and turbulent heat flux and is an important driver of the SMB, which should be analysed aswell.

Given that the spatial distribution of surface mass balance and albedo from FAMOUS-ice differ strongly from respective fields simulated by MAR, I would propose to additionally show selected surface exchange characteristics (Fig. 2) as 2-D fields in comparison to MAR.

It is important to not only analyse the spatial characteristics of the simulated SMB but to also evaluate the model's response to climate variations. The model is explicitly designed to simulate fundamental climate change and consequently this paper should allow to assess the model's skill to respond to climate change. This could be accomplished by using a transient ocean forcing from a 21st century climate projection.

Furthermore, I found the manuscript at times hard to read, as essential information with respect to the snow pack schemes are only found in referenced literature:

Page 10: It would be helpful to briefly summarize which processes influence the grain-size of snow and to generally characterize the relationship between snow albedo and climate forcing, and to give typical grain sizes of new snow and old/wet snow. (Alternatively, a figure illustrating albedo as a function of influencing factors could be beneficial.)

This is particularly desirable as Marshall (1989) is not easily available and as Gardner and Sharp (2010) indicates a sensitivity to grain size $\approx 0.05$ / mm towards greater grain sizes, which seems to be considerably smaller than the values used here.

I also missed a separate short paragraph on how the snow pack evolvels over time: How does density change if snow accumulates on top, what happens if the first layer melts, how and when are densities of the snow column prognostically calculated, what is the typical depth of compacted snow?

I also recommend to improve the structure of the paper by a further break down of subsections. Separate subsections might cover initial conditions, upper and lower boundary conditions of the snow pack as seen by the land surface model, and boundary conditions at the ice surface as seen from the atmosphere. Finally the analyzed experiments, together with the MIROC forcing and the MAR and RACMO simulation should be introduced in a separate section, possibly before section 4.

**3 Some specific comments**

p. 5, l. 8: Is it possible to quantify the computational cost of the new model version in comparison to the old version?
p. 5, l. 17: "We will describe later..." Please refer to the respective section.
p. 7, l. 9: high-latitude -> high-altitude?
p. 7, l. 11: Is the sub-surface lapse rate in line with the coarse resolution subsurface temperature distribution?
p. 11, l. 12: (fig:4) -> (Fig. 2) ?
Fig. 3, 5,6, : As the colorbar does not caver the full range of values, maybe inlude the range of values in the figure caption.
p. 12-14: does the spatial representation of SMB, albedo improve qualitatively if the number of elevation classes is increased?

p. 12, l. 9: It should be stressed that FAMOUS and MAR are consistently forced with the climate model output while RACMO is forced by reanalysis data. Also please specify the ERA forcing used and the period which was analysed.

p. 14, Tab. 1: Please specify which ERA forcing is used here, Fettweis 2013 only has RACMO(ECMWF), and the RACMO(ERA) experiment that was used in this paper should be added to this table.

**References**

Fettweis, X., Hofer, S., Krebs-Kanzow, U., Amory, C., Aoki, T., Berends, C. J., Born, A., Box, J. E., Delhasse, A., Fujita, K., Gierz, P., Goelzer, H., Hanna, E., Hashimoto, A., Huybrechts, P., Kapsch, M.-L., King, M. D., Kittel, C., Lang, C., Langen, P. L., Lenaerts, J. T. M., Liston, G. E., Lohmann, G., Mernild, S. H., Mikolajewicz, U., Modali, K., Mottram, R. H., Niwano, M., Noël, B., Ryan, J. C., Smith, A., Streffing, J., Tedesco, M., van de Berg, W. J., van den Broeke, M., van de Wal, R. S. W., van Kampenhout, L., Wilton, D., Wouters, B., Ziemen, F., and Zolles, T.: GrSMBMIP: Intercomparison of the modelled 1980–2012 surface mass balance over the Greenland Ice sheet, The Cryosphere Discussions, 2020, 1–35, https://doi.org/10.5194/tc-2019-321, https://www.the-cryosphere-discuss.net/tc-2019-321/, 2020.

Gardner, A. S. and Sharp, M. J.: A review of snow and ice albedo and the development of a new physically based broadband albedo parameterization, Journal of Geophysical Research: Earth Surface, 115, https://doi.org/10.1029/2009JF001444, https://agupubs.onlinelibrary.wiley.com/doi/abs/10.1029/2009JF001444, 2010.

Kapsch, M.-L., Mikolajewicz, U., Ziemen, F. A., Rodehacke, C. B., and Schannwell, C.: Analysis of the Surface Mass Balance for Deglacial Climate Simulations, The Cryosphere Discussions, 2020, 1–40, https://doi.org/10.5194/tc-2020-173, https://tc.copernicus.org/preprints/tc-2020-173/, 2020.

---

## Referee Comment (RC2) · Anonymous Referee #2 · 8 Sep 2020

**General comments**

The earth system model FAMOUS-ice, which now includes an updated multi-layer snow scheme and coupled downscaling of the surface mass balance using elevation tiles, is presented in the manuscript. Synchronous coupled ice sheet-climate simulations is an active area of research, and this manuscript is a timely contribution. The computational efficiency makes FAMOUS-ice well-suited and capable for long climate simulations, where the evolution of the ice sheets play a crucial role.

FAMOUS-ice is not capable of accurately resolving orographic precipitation over the ice sheet, and is biased warm at high altitudes. The resulting SMB has obvious spatial

differences with the regional climate model MAR simulated SMB. In particular, surface melt occurs at too high altitudes due to a low albedo. Due to this, the manuscript could be improved by adding a more detailed analysis of the surface energy balance.

Before publication I recommend minor revisions.

**Scientific comments**

You argue that Helsen et al., 2012 developed an empirical parameterization to translate global climate fields to usable boundary conditions. However, they state that their method is not usable for models where ablation areas are not resolved.

How does FAMOUS-ice prevent snow from growing infinitely thick? Does it include a firn-to-ice densification scheme, or do you cap the snow above a certain thickness?

For downscaling of the temperature and (incoming?) longwave radiation from the atmosphere to the elevation classes you use an empirically tuned method, where you assume that the near-surface climate gradients are similar to the climate gradients in the lower atmosphere. Is this a valid assumption? For example, the temperature decreases with elevation along the GrIS near-surface, while the lower atmosphere has a temperature increase with height due to inversion.

I am not convinced you improve on the elevation class implementation analysed by Sellevold et al., 2019, as you are not comparing the same metric - that is, you do not subtract the grid-cell average from the same grid-box elevation class simulated value.

Below 1000 m, the latent heat flux is positive in FAMOUS (Fig. 2), while negative in MAR and RACMO. First, it should be indicated whether the positive values means energy loss or gain at the surface. I am assuming the latter, as MAR and RACMO simulate climatological mean sublimation at lower elevations. Can you explain why FAMOUS simulates climatological mean deposition at these elevations?

You argue that the shortwave down is too low because of a high biased cloud cover. I am surprised to see that this is not compensated for by an overestimation of longwave

down. Is this due to the atmosphere being warmer in FAMOUS?

The simulation of albedo shows clear differences, particularly at higher elevations, when compared to MAR. You argue that the trigger for the lowered albedo at the higher elevations is a warm temperature bias at higher altitudes. Including a (supplementary) figure of April/May near-surface temperature or snow grain size compared to MAR/observations would likely strengthen this argument. Further, is it possible to assess whether the albedo is triggered too much and/or too early in the melt season?

Also, if the atmospheric temperatures are too high at high altitudes, the melt energy contribution from the sensible heat flux should also be higher. As you don't show the sensible heat flux, I am left to wonder how the simulation of the sensible heat flux is.

How well is rain simulated over the ice sheet? As the ice sheet is warm at higher altitudes, is there an overestimation of rainfall at higher altitudes? Could this also contribute to a lower albedo?

The presented surface mass balance shows a realistic distribution with ablation at lower elevations and accumulation at higher elevations. However, the ablation areas are biased large and the high accumulation areas in the Southeast and Northwest show too little accumulation. As you don't present a transient climate simulation, I think the possible effect of these biases in transient simulations deserves a paragraph in the discussion section.

It would be easier to understand some of the thickness anomalies in Fig. 8 if you showed the ice velocities. Consider adding it to supplementary materials.

**Technical comments**

I found many occurrences of double parentheses with citations. If latex is used, use (text before citation ) or  to avoid.

P. 11, l. 12: replace fig:4 with figure 4.

P. 19, l. 2: "a model"

P. 19, l 2: "allow for coupled"

Fig. 7, caption: replace x in: 10x km3

────────────────────────────

---

## Author Comment (AC1) · 23 Oct 2020

**Response to reviews**

We would like to thank both reviewers for their thoughtful comments on our paper. We were glad to see that both thought a revised manuscript would make a timely and important contribution to the field as a full GMD paper. We agree that additional analysis and content along the lines they suggest will improve our model description - these revisions will take some time to

5 complete, and in the meantime, this point-by-point response will address the comments they made and outline how we intend to revise the paper.

As written, our paper was intended primarily as a technical description of how our novel FAMOUS-to-ice-sheet coupling has been implemented, with a focus on the downscaling techniques and enough illustrative detail in the results to show that our coupled model simulations could be scientifically plausible even with the low native resolution of the FAMOUS atmosphere.

- 10 We are pleased to see that Reviewer 1 thinks we have succeeded in this aim, at least for a modern climate state (first blue sentence below). Given this intention, we are not keen on having this paper include extensive analysis and evaluation of any individual climate simulation where that would not speak to our primary technical focus. Certain issues that significantly affect the setup of a simulation with a coupled climate-ice model (for instance spinup/initialisation techniques, which are still the subject of much ongoing research) we considered to be out of scope for this paper. It's clearly not sensible to suggest
- 15 that a modelling technique should be described without giving some evidence that its results are likely to be fit for some stated purpose if applied appropriately, and the results and analysis we have done - and will expand on in a revised manuscript, guided by the review comments - are framed with this in mind. We wish to demonstrate the capabilities of the coupling techniques in the model, rather than evaluate the large-scale climate of the FAMOUS atmosphere or the general behaviour of the Glimmer ice sheet (both of which have been done in previous studies) under a particular set of boundary conditions. Our responses
- 20 below will note where we agree with an additional analysis suggested by the reviewers, and also where we do not think that what they suggest will usefully serve the purpose we have in mind for this paper.

Below, the reviewers' comments are in blue italics and our responses to each follow in black.

**Reviewer 1**

45

The presented analysis demonstrates that FAMOUS-ice in combination with an ice sheet model is capable to simulate a reasonably realistic GrIS under present day climate conditions. The surface mass balance, however, appears to be extended to too high elevations and albedo seems to be overestimated in the ice sheet's interior. Given that other ESMs of similar design and resolution exhibit a qualitatively better skill (e.g. Kapsch et al., 2020; Fettweis et al., 2020), the analysis of the model should answer the question, whether these shortcomings are the result of a biased climate or of a misrepresentation of specific processes at the ice sheet's surface.

- 30 It is true of course that the simulation of surface mass balance components in FAMOUS-ice contains biases. We would note that the comparison with the ESMs contributing to GrSMBMIP (Fettweis et al., 2020) is perhaps too stringent a test for FAMOUS-ice both CESM2 and MPI-ESM are much more physically complex and computationally expensive CMIP6-class models run at significantly higher resolution (CESM2's GrSMBIP submission was from a 1 degree model (37 gridboxes for every 1 in FAMOUS) and 32 vertical levels (almost 3x as many as FAMOUS) and MPI-ESM submitted results from an even
  35 bigher resolution model T127\_05 vartical levels
- 35 higher resolution model T127, 95 vertical levels.

Regardless of what is used as a comparison or its computational cost, FAMOUS-ice must be shown to stand on its own merits as suitable for its intended purpose, which we think it does - it seems Reviewer one agrees (their first sentence). Their question of whether the SMB, albedo and other GrIS biases that we do have are rooted in the large-scale climate or our downscaled surface modelling is a good one though, and very relevant to what we want to address. We will discuss this topic further in the

40 revised paper. It's unlikely to be a straightforward either/or question, and shortcomings in both will play different roles. It's also unlikely to be just a question of misrepresented processes, as the resolution in FAMOUS is going to have a significant part on its own.

A very similar approach to ice sheet surface mass balance - similar down to the code level - is taken in UKESM1 (Sellar et al. 2019, a paper with significant additional description and evaluation of the ice surface schemes is in preparation) at a much higher resolution, equivalent to the CESM2 configuration noted above. GrIS surface fields in UKESM1 are generally much

1

more similar to those in RCMs (eg MAR) than the FAMOUS fields we use to illustrate this paper, which suggests that the process parameterisations in our surface scheme are fundamentally capable of doing better, but are being held back by other factors in FAMOUS-ice.

Accumulation is not downscaled to elevation tiles in FAMOUS-ice, and is a pure output of the 5x7.5 degree atmosphere.

- 5 Global GCM atmospheres are well-known for producing widespread, persistent drizzle where they should be dry even in modern, higher resolution models, and the smoothed, relatively low orography of GrIS and relatively high levels of moisture diffusivity inherent to the low resolution FAMOUS grid exacerbate this tendency. The significant bias in downwelling short-wave radiation over GrIS shown in our Figure 2 is also a product of the FAMOUS atmosphere, and can't be directly attributed to failings in the surface modelling. One might assume that the low albedo biases in Figure 2 and 3 would be a clear case of
- 10 shortcomings in the surface scheme, but in fact these are partly result from tuning to compensate for the downwelling shortwave bias, so even the albedo bias is not simple to attribute.

15

Ultimately, to produce a "good" set of surface fields from a seriously biased background climate would require significant, deliberately unphysical interventions, such as using flux adjustments. That would not be an appropriate strategy for a model such as ours, intended for use in climate very different from modern, observed ones, and a balance must be struck between accepting the biases that are present and some model tuning that makes allowances for them.

Fig. 2 indicates that the downward shortwave radiation is strongly biased towards lower values while downward longwave radiation does not exhibit any clear bias. As argued by the authors, this may indicate an overestimated cloud cover, which should be substantiated by decomposing long wave radiation into atmospheric emissivity and near surface temperature. A potentially biased cloud cover should be reflected in atmospheric emissivity. Near surface temperature influences both, downward

- 20 longwave radiation and turbulent heat flux and is an important driver of the SMB, which should be analysed as well. Given that the spatial distribution of surface mass balance and albedo from FAMOUS-ice differ strongly from respective fields simulated by MAR, I would propose to additionally show selected surface exchange characteristics (Fig. 2) as 2-D fields in comparison to MAR. It is important to not only analyse the spatial characteristics of the simulated SMB but to also evaluate the model's response to climate variations. The model is explicitly designed to simulate fundamental climate change and consequently this
- 25 paper should allow to assess the model's skill to respond to climate change. This could be accomplished by using a transient ocean forcing from a 21st century climate projection.

Reviewer 2 also queries how the cloud is affecting both short- and longwave components of the radiation forcing, and we will look further to explain this for the revision and see how the downscaling that is applied to downwelling longwave in each gridbox features in this. These radiation components are also relevant to our scheme's modelling of albedo, so they are relevant

30 from the point of view of our technical focus. As suggested, we will produce additional plots of the spatial variation of surface characteristics to complement the line plots in figure 2, and disentangle the effects of the elevation downscaling from simple variation with height along the mean ice sheet surface as suggested.

The sensitivity of the coupling scheme to climate change in FAMOUS is a matter raised by both reviewers. This is clearly an important topic, given the intended use of the coupled model, and we will include some analysis of it in the revised paper.

- 35 However, given the caveats towards general simulation evaluation given in our preamble we are wary of going too far down this path in this paper since the SMB response will be controlled to a large degree by the climatic of FAMOUS to a specific forcing. We note that Gregory et al (2020, currently under revision for TC (tc-2020-89), first version available in TCD, GGS20 hereafter) already explores in some depth the response of an ensemble of the coupled FAMOUS-ice Glimmer system to a range of climate change forcings, so our additions will take GGS20 as context and provide complementary detail. GGS20 shows that
- 40 GrIS SMB climate sensitivity of FAMOUS-ice is a reasonable match to the polynomial scaling developed by Fettweis et al. 2013 and used in AR4.

Rather than use a transient ocean forcing as suggested by the reviews, for our revision we propose to compare detail from one of the GGS20 ensemble, forced continuously by end of 21st century MIROC RCP4.5 SSTs, with MAR results diagnosed at the end of the 21st century from their transient forcing with the MIROC RCP4.5 projection. We note that this will be presented

45 as a comparison between model projections, rather than assessment of "skill" as suggested by Reviewer 1. We don't consider that it is possible to assess a model's skill in the context of a climate projection, for which there is no verifiable truth (yet).

Furthermore, I found the manuscript at times hard to read, as essential information with respect to the snow pack schemes are only found in referenced literature: Page 10: It would be helpful to briefly summarize which processes influence the grainsize of snow and to generally characterize the relationship between snow albedo and climate forcing, and to give typical grain sizes of new snow and old/wet snow. (Alternatively, a figure illustrating albedo as a function of influencing factors could be beneficial.) This is particularly desirable as Marshall (1989) is not easily available and as Gardner and Sharp (2010) indicates a sensitivity to grain size (0.05 /mm) towards greater grain sizes, which seems to be considerably smaller than the values used here

5 We will provide more background, and summarise factors that influence snow albedo in the revised paper. Snow grain size distributions and seasonal evolution will be included in the set of fields analysed to characterise the capability of the model and explain the resulting albedo fields.

To address the particular comment on the sensitivity of the albedo to snow grain size: improving our match to observations of this relationship was the primary tuning criteria used in changing the parameters in our scheme. Compared to the original parameters in this version of the snow scheme in JULES, our model has an increased sensitivity to grain size (figure R1).

The changes were made for the visible part of the two-stream radiation scheme in FAMOUS. Snow grain size is limited to  $2000\mu$ m, which leads to an associated minimum in albedo. These adjustments were made to provide a more realistic representation of seasonal snow albedo evolution (e.g., Stroeve (2013) https://doi.org/10.1016/j.rse.2013.07.023). Additional observational studies were referenced, as was work on albedo optimisation in two-stream radiation code in ECHAM4 (Roesch(2012),

10.1029/2001JD000809). 15

10

The tuning parameter for which values are given in the main text,  $\Delta a_{\text{snow,visible}}$ , is not directly used as the sensitivity of albedo to the grain size, but as part of a larger formulation. It is thus not directly comparable to the 0.05 /mm inferred from Gardner and Sharp (2010) by the reviewer. As can be seen from the orange line in figure R1, the albedo sensitivity to grain size that results from our formulation is in fact similar to this 0.05 /mm estimate, although the gradient is not constant across the

20 full range of grainsize.

Figure R1. Snow albedo changes with grain size. Blue line shows the original tuning of the scheme in FAMOUS-ice, the orange line is the Figure R2. Seasonal maximum grain size occurring somewhere in result of our tuning to observational studies

FAMOUS-ice. Each line represents a different sample year. Red lines are for snow on non-ice surfaces, blue lines are for snow on the surface of the main ice sheet

As can be seen in figure R2, grain size for surface snow on the main body of the ice sheet often reaches the maximum value possible in one or more places during the summer melt season. This will lead to the lowest possible snow albedos in our scheme for these surfaces. This figure does not show influence of bare ice or near-surface air temperatures high enough to cause melt that will lower albedo further. These low albedos do not persist year round, and by the end of the winter fresh snow has covered over the higher grain-size snow. The low albedos shown in figure 3 of the main text are a summer phenomenon only - summer is when the albedo is most important, of course!

I also missed a separate short paragraph on how the snow pack evolves over time: How does density change if snow accumulates on top, what happens if the first layer melts, how and when are densities of the snow column prognostically calculated, what is the typical depth of compacted snow?

As requested, the revision will include more description of the snow relayering process, to save the need to refer further back in the literature

**5**

I also recommend to improve the structure of the paper by a further break down of sub-sections. Separate subsections might cover initial conditions, upper and lower boundary conditions of the snow pack as seen by the land surface model, and boundary conditions at the ice surface as seen from the atmosphere. Finally the analyzed experiments, together with the MIROC forcing and the MAR and RACMO simulation should be introduced in a separate section, possibly before section 4

- We have no objection to reconsidering the subdivision of material in the paper. However, as noted already, we would like 10 the focus to remain on a technical description of the methods used and the general capabilities of the model rather than a detailed evaluation of one particular climate realisation. Some of the suggestions made in this comment would not fit with our technical focus - for instance, description of initial conditions for the simulation. The choice of appropriate initial conditions can depend on the purpose of a specific experiment, so is not clearly part of a description of general model capabilities. GGS20
- describe and justify the spinup strategy used for the experiments presented there. Once the additional analysis described above 15 is complete, we will reconsider the structure of the paper

p. 5. I. 8: Is it possible to quantify the computational cost of the new model version in comparison to the old version?

ves, this will be noted. The additional tiles do have a non-negligible impact on the cost of FAMOUS, although appropriately configured on 10 processors throughput can still be measured in 100s of simulated years per wallclock day.

p. 5, l. 17: "We will describe later..." Please refer to the respective section. 20

will do

p. 7, l. 9: high-latitude -> high-altitude?

either would fit, I think, for this example. By "high-latitude" we meant latitudes near the poles

p. 7, l. 11: Is the sub-surface lapse rate in line with the coarse resolution subsurface temperature distribution?

That's a good question. In places is looks like it is, on this metric other transects across the ice sub-surface would suggest a 25 higher lapse rate might be more appropriate. It's not, in fact a value we have tried to tune, so we don't know what the sensitivity of our results to it is. Other experimentation with the ice sub-surface formulation during development suggests that this lapse rate would not be expected to have a major effect on our results.

p. 11, l. 12: (fig:4) -> (Fig. 2) ?

30 yes, that's a typo.

> Fig. 3, 5,6, : As the colorbar does not cover the full range of values, maybe include the range of values in the figure caption. good idea

p. 12-14: does the spatial representation of SMB, albedo improve qualitatively if the number of elevation classes is increased?

- The number of elevation classes used in a full simulation has not been changed since initial tests at the start of model 35 development, and further simulations to assess how sensitive the SMB simulation is to this factor alone would not be straightforward. In an early version of FAMOUS-ice we used 25 evenly-spaced levels, and found that results with the same 10 class level placement as CESM1 used produced equivalent large-scale results at less computational cost. All further snow model developments and tuning were then done with these 10 classes as a base.
- 40 On the evidence we have I don't think it's likely that additional classes, with potential retuning of the surface parameters, would make a major difference to our results or the biases present in the surface fields - see discussion above of whether surface processes or large-scale climate are more responsible for the biases we see. Those background climate factors would not change with increased numbers of elevation classes at the surface.

p. 12, l. 9: It should be stressed that FAMOUS and MAR are consistently forced with the climate model output while RACMO is forced by reanalysis data. Also please specify the ERA forcing used and the period which was analysed. 45 will do

p. 14, Tab. 1: Please specify which ERA forcing is used here, Fettweis 2013 only has RACMO(ECMWF), and the RACMO(ERA) experiment that was used in this paper should be added to this table will do

**Reviewer 2**

30

35

45

You argue that Helsen et al., 2012 developed an empirical parameterization to translate global climate fields to usable boundary conditions. However, they state that their method is not usable for models where ablation areas are not resolved.

OK, this will be rephrased.

5 *How does FAMOUS-ice prevent snow from growing infinitely thick? Does it include a firn-to-ice densification scheme, or do you cap the snow above a certain thickness?*

The snow scheme does include snowpack compaction and densification processes, although over time this alone would not prevent infinite columns of ice-density "snow" from forming where local accumulation is greater than ablation. Indeed, when not coupled to an ice sheet model, all UM configurations old and modern that we are aware of do slowly accumulate very large

- 10 snow masses in such regions until someone thinks to reset the initial conditions this is a numerical quirk that has very little effect on the climate simulation due to the way that the snow surface is implemented in these configurations. In FAMOUS-ice we use the coupling scheme as outlined in section 8.1, and the snowpack mass is reset to a default value at the start of every year. The amount of mass added/subtracted in the course of this resetting is what we pass to the ice sheet as the SMB term. In the revised paper this will be clarified as part of the new paragraph requested by reviewer 1 that will describe the relayering of
- 15 the evolving state of the snowpack.

For downscaling of the temperature and (incoming?) longwave radiation from the atmosphere to the elevation classes you use an empirically tuned method, where you assume that the near-surface climate gradients are similar to the climate gradients in the lower atmosphere. Is this a valid assumption? For example, the temperature decreases with elevation along the GrIS near-surface, while the lower atmosphere has a temperature increase with height due to inversion.

- 20 The surface temperature and downwelling longwave are downscaled with elevation using a fixed lapse rate. This is commonly done in this sort of downscaling (see references in the original paper), although it is undeniably a rather simplistic method and cannot reflect local conditions such as inversions that deviate from its cooling-with-height paradigm. The value used for this lapse rate is sometimes treated as a tunable parameter in order to achieve a realistic surface temperature distribution on the ice sheet. Used, as we do, to reflect subgrid-scale surface variation in a single land model gridbox, this near-surface-like scaling is
- 25 more justified than trying to use the state of the free atmosphere column, especially in a lower-resolution model like FAMOUS where the vertical resolution in the atmosphere is rather low. As noted in the text, we did try this latter approach initially but found it did not produce good results in FAMOUS. Some of this discussion will be included in the revised text.

I am not convinced you improve on the elevation class implementation analysed by Sellevold et al., 2019, as you are not comparing the same metric - that is, you do not subtract the grid-cell average from the same grid-box elevation class simulated value.

My impression of the CESM1 classes shown in Sellevold et al. (2019) was that several quantities were found, per gridbox, to increase with height where they should decrease and vice versa (their Figures 1,2) whereas our Figures 2 and 4 imply that overall we reproduce relationships with height in the correct sense for all variables. I do accept that their metric is more detailed, and perhaps more of a stringent test than what we have shown here. We will attempt to reproduce their metric more precisely and amend the revised manuscript accordingly.

Below 1000 m, the latent heat flux is positive in FAMOUS (Fig. 2), while negative in MAR and RACMO. First, it should be indicated whether the positive values means energy loss or gain at the surface. I am assuming the latter, as MAR and RACMO simulate climatological mean sublimation at lower elevations. Can you explain why FAMOUS simulates climatological mean deposition at these elevations?

40 Thanks for this - there was a sign error in converting the FAMOUS-ice sublimation diagnostic into latent heat in this plot. There is indeed sublimation at lower elevations, and low levels of deposition higher up. Latent heat fluxes on Greenland in FAMOUS-ice are generally very small: looking at the spatial patterns of the surface fluxes in the revision, as requested by Reviewer 1, will help clarify what is going on.

You argue that the shortwave down is too low because of a high biased cloud cover. I am surprised to see that this is not compensated for by an overestimation of longwave down. Is this due to the atmosphere being warmer in FAMOUS?

Reviewer 1 also requested more analysis of how the cloud and radiation components produce the surface energy balance we see: this will be addressed in the revised paper.

The simulation of albedo shows clear differences, particularly at higher elevations, when compared to MAR. You argue that the trigger for the lowered albedo at the higher elevations is a warm temperature bias at higher altitudes. Including a (supplementary) figure of April/May near-surface temperature or snow grain size compared to MAR/observations would likely strengthen this argument. Further, is it possible to assess whether the albedo is triggered too much and/or too early in the melt season?

5 season

25

As requested by reviewer 1, a number of additional fields comparing the 2d surface state of FAMOUS-ice with MAR. The low simulation of albedo is a melt season phenomenon - winter albedo, dominated by the low grain size of freshly fallen snow, is generally high everywhere. Figure R2 shows that grain size for surface snow often reaches maximum values on the ice sheet during the summer. There is a feedback between higher melt, lower albedo and faster-evolving grainsize, so it is not trivial to

10 clearly identify a single initial trigger. We will show more analysis of the causes and timing of the albedo we simulate in the revised paper.

Also, if the atmospheric temperatures are too high at high altitudes, the melt energy contribution from the sensible heat flux should also be higher. As you don't show the sensible heat flux, I am left to wonder how the simulation of the sensible heat flux is.

this will be included as part of the additional comparison of the surface state compared with MAR

How well is rain simulated over the ice sheet? As the ice sheet is warm at higher altitudes, is there an overestimation of rainfall at higher altitudes? Could this also contribute to a lower albedo?

There is an over-estimation of low-intensity rain. In common with most GCMs, especially lower resolution ones, FAMOUS tends to drizzle too much everywhere. However, rain is ignored by the snow pack model in FAMOUS-ice - it isn't allowed to

20 pool on top or percolate through the snowpack, but instead runs off straight away to the ocean. It has no direct effect on the albedo calculation or snow pack properties in general

The presented surface mass balance shows a realistic distribution with ablation at lower elevations and accumulation at higher elevations. However, the ablation areas are biased large and the high accumulation areas in the Southeast and Northwest show too little accumulation. As you don't present a transient climate simulation, I think the possible effect of these biases in transient simulations deserves a paragraph in the discussion section.

Reviewer 1 also requested some consideration of how the model responds to climate change, this is discussed above. We note again that GGS20 shows that the sensitivity of GrIS SMB to climate change forcing in FAMOUS-ice is similar to the polynomial scaling estimated by Fettweis et al. (2013) using MAR simulations. Some analysis of the SMB and surface exchange fluxes in FAMOUS-ice under climate change will be considered in the revised manuscript.

30 It would be easier to understand some of the thickness anomalies in Fig. 8 if you showed the ice velocities. Consider adding it to supplementary materials.

this will be done

I found many occurrences of double parentheses with citations. If latex is used, use(text before citation) or to avoid. we'll fix this

35 *P. 11, l. 12: replace fig:4 with figure 4.* will do *P. 19, l. 2: "a model"*

will fix

P. 19, l 2: "allow for coupled"

I think this sentence is actually grammatically OK as it stands?
 *Fig. 7, caption: replace x in: 10x km* will do

---

## Author Response (AR1)

**Response to reviews, edited from the GMDD author comment to accompany resubmitted ms.**

We would like to thank both reviewers for their thoughtful comments on our paper. We were glad to see that both thought a revised manuscript would make a timely and important contribution to the field as a full GMD paper. We agree that additional analysis and content along the lines they suggest would improve our model description, and here note in detail changes we have made in response to each point raised.

As written, our paper was intended primarily as a technical description of how our novel FAMOUS-to-ice-sheet coupling has been implemented, with a focus on the downscaling techniques and enough illustrative detail in the results to show that our coupled model simulations could be scientifically plausible even with the low native resolution of the FAMOUS atmosphere. We are pleased to see that Reviewer 1 thinks we have succeeded in this aim, at least for a modern climate state (first blue sentence below). Given this intention, we are not keen on having this paper include extensive analysis and evaluation of any individual climate simulation where that would not speak to our primary technical focus. Certain issues that significantly affect the setup of a simulation with a coupled climate-ice model (for instance spinup/initialisation techniques, which are still the subject of much ongoing research) we considered to be out of scope for this paper. It's clearly not sensible to suggest that a modelling technique should be described without giving some evidence that its results are likely to be fit for some stated purpose if applied appropriately, and the results and analysis we have done are framed with this in mind. We wish to demonstrate the capabilities of the coupling techniques in the model and describe typical results, rather than evaluate the large-scale climate of the FAMOUS atmosphere or the general behaviour of the Glimmer ice sheet (both of which have been done in previous studies) under any particular set of boundary conditions.

Below, the reviewers' comments are in blue italics and our responses to each follow in black. Line numbers given in our responses refer to the track-changed version of the resubmitted manuscript, line numbers in reviewer comments refer to the original discussion paper.

**Reviewer 1**

*The presented analysis demonstrates that FAMOUS-ice in combination with an ice sheet model is capable to simulate a reasonably realistic GrIS under present day climate conditions. The surface mass balance, however, appears to be extended to too high elevations and albedo seems to be overestimated in the ice sheet's interior. Given that other ESMs of similar design and resolution exhibit a qualitatively better skill (e.g. Kapsch et al., 2020; Fettweis et al., 2020), the analysis of the model should answer the question, whether these shortcomings are the result of a biased climate or of a misrepresentation of specific processes at the ice sheet's surface.*

It is true of course that the simulation of surface mass balance components in FAMOUS-ice contains biases. We would note that the comparison with the ESMs contributing to GrSMBMIP (Fettweis et al., 2020) is perhaps too stringent a test for FAMOUS-ice - both CESM2 and MPI-ESM are much more physically complex and computationally expensive CMIP6-class models run at significantly higher resolution (CESM2's GrSMBIP submission was from a 1 degree model (37 gridboxes for every 1 in FAMOUS) and 32 vertical levels (almost 3x as many as FAMOUS) and MPI-ESM submitted results from an even higher resolution model - T127, 95 vertical levels.

Regardless of what is used as a comparison or its computational cost, FAMOUS-ice must be shown to stand on its own merits as suitable for its intended purpose, which we think it does - it seems Reviewer one agrees (their first sentence). Their question of whether the SMB, albedo and other GrIS biases that we do have are rooted in the large-scale climate or our downscaled surface modelling is a good one though, and very relevant to what we want to address. This is now addressed in a substantial new paragraph in the Discussion (page 23).

*Fig. 2 indicates that the downward shortwave radiation is strongly biased towards lower values while downward longwave radiation does not exhibit any clear bias. As argued by the authors, this may indicate an overestimated cloud cover, which should be substantiated by decomposing long wave radiation into atmospheric emissivity and near surface temperature. A potentially biased cloud cover should be reflected in atmospheric emissivity. Near surface temperature influences both, downward longwave radiation and turbulent heat flux and is an important driver of the SMB, which should be analysed as well. Given that the spatial distribution of surface mass balance and albedo from FAMOUS-ice differ strongly from respective fields simulated*

*by MAR, I would propose to additionally show selected surface exchange characteristics (Fig. 2) as 2-D fields in comparison to MAR. It is important to not only analyse the spatial characteristics of the simulated SMB but to also evaluate the model's response to climate variations. The model is explicitly designed to simulate fundamental climate change and consequently this paper should allow to assess the model's skill to respond to climate change. This could be accomplished by using a transient*
5  *ocean forcing from a 21st century climate projection.*

For the modern simulation originally addressed in our paper, there is now more discussion of the downscaled surface fluxes, including turbulent fluxes, starting on page 7, line 29, and 2D spatial distributions of a number of relevant fields are now included as Supplementary material. Downwelling longwave is now plotted for JJA for easier comparison to the shortwave, and a compensation between the two radiative components can simply be seen for the FAMOUS-ice bias wrt MAR - see also
10  Reviewer 2's comments on this. The scope of our analysis is somewhat limited by the availability of comparable diagnostics from the model simulations in question. The original FAMOUS-ice simulations used to illustrate this paper did not output usable diagnostics of sensible heat flux, so new ones have been run to provide this field. By necessity these were conducted on a different computing platform and are not perfect reproductions of the earlier simulations. For this reason, some of the minor detail in the FAMOUS-ice figures has changed in this revision, as have the FAMOUS-ice data in Table 1. The 5-letter code
15  (xotzb) used in the title to denote the precise configuration of FAMOUS used has also changed to match.

The sensitivity of the coupling scheme to climate change in FAMOUS is a matter raised by both reviewers. This is clearly an important topic given the intended use of the coupled model, however given the caveats towards general simulation evaluation given in our preamble we are wary of analysing the model climate too far down this path in this paper. The SMB response will be controlled to a large degree by the background climate response of FAMOUS to climate warming, which has been studied
20  previously in the literature. We note that Gregory et al. (The Cryosphere, 2020) already explores in some depth the response of an ensemble of the coupled FAMOUS-ice Glimmer system to a range of climate change forcings, so our additions will take that paper as context and provide complementary detail. Consequently, taking one future climate scenario as illustrative, we have created a new subsection (7.2, page 17) looking at how some of the SMB component fields change under a warmer climate, and added new data to Table 1 of integrated SMB and its components (page 17).

25  *Furthermore, I found the manuscript at times hard to read,as essential information with respect to the snow pack schemes are only found in referenced literature:Page 10: It would be helpful to briefly summarize which processes influence the grain-size of snow and to generally characterize the relationship between snow albedo and climate forcing, and to give typical grain sizes of new snow and old/wet snow. (Alternatively, a figure illustrating albedo as a function of influencing factors could be beneficial.) This is particularly desirable as Marshall (1989) is not easily available and as Gardner and Sharp (2010) indicates*
30  *a sensitivity to grain size (0.05 /mm) towards greater grain sizes, which seems to be considerably smaller than the values used here*

Influences on the grain-size of snow and what happens in our snow model are now described on page 9, line 31, and a new figure 3 shows how our resultant broadband snow albedo varies as a function of grain-size. Compared to the original parameters in this version of the snow scheme in JULES, our model has an increased sensitivity to grain size (new figure 3).
35  The changes were made for the visible part of the two-stream radiation scheme in FAMOUS. Snow grain size is limited to $2000\mu$m, which leads to an associated minimum in albedo. These adjustments were made to provide a more realistic representation of seasonal snow albedo evolution (e.g., Stroeve (2013) https://doi.org/10.1016/j.rse.2013.07.023). Additional observational studies were referenced, as was work on albedo optimisation in two-stream radiation code in ECHAM4 (Roesch(2012), 10.1029/2001JD000809). The tuning parameter for which values are given in the main text, $\Delta a_{\text{snow,visible}}$, is not directly used
40  as the sensitivity of albedo to the grain size, but as part of a larger formulation. It is thus not directly comparable to the 0.05 /mm inferred from Gardner and Sharp (2010) by the reviewer. As can be seen from the orange line in the new figure 3, the albedo sensitivity to grain size that results from our formulation is in fact similar to this 0.05 /mm estimate, although the gradient is not constant across the full range of grainsize.

*I also missed a separate short paragraph on how the snow pack evolves over time:How does density change if snow accu-*
45  *mulates on top, what happens if the first layer melts, how and when are densities of the snow column prognostically calculated, what is the typical depth of compacted snow?*

There is now a paragraph on page 9 (starts line 7) describing the evolution and relayering of the snow pack model as mass is added or removed.

*I also recommend to improve the structure of the paper by a further break down of sub-sections. Separate subsections might cover initial conditions, upper and lower boundary conditions of the snow pack as seen by the land surface model, and boundary conditions at the ice surface as seen from the atmosphere. Finally the analyzed experiments, together with the MIROC forcing and the MAR and RACMO simulation should be introduced in a separate section, possibly before section 4*

We have no objection to reconsidering the subdivision of material in the paper. However, as noted already, we would like the focus to remain on a technical description of the methods used and the general capabilities of the model rather than a detailed evaluation of one particular climate realisation. Some of the suggestions made in this comment would not fit with our technical focus - for instance, description of initial conditions for the simulation, and on reflection we have decided to keep the structure much as it was.

*p. 5, l. 8: Is it possible to quantify the computational cost of the new model version in comparison to the old version?*

Computational expense of this version of FAMOUS are now in the Discussion, page 24, line 23.

*p. 5, l. 17: "We will describe later..." Please refer to the respective section.*

done (here and elsewhere)

*p. 7, l. 9: high-latitude -> high-altitude?*

either would fit, I think, for this example. By "high-latitude" we meant latitudes near the poles

*p. 7, l. 11: Is the sub-surface lapse rate in line with the coarse resolution subsurface temperature distribution?*

That's a good question. This is noted on page 7, line 20

*p. 11, l. 12: (fig:4) -> (Fig. 2) ?*

yes, that's a typo.

*Fig. 3, 5,6, : As the colorbar does not cover the full range of values, maybe include the range of values in the figure caption.*

good idea, this has been done for all figures with spatial distributions in

*p. 12-14: does the spatial representation of SMB, albedo improve qualitatively if the number of elevation classes is increased?*

This is discussed now in the Discussion on page 23, line 29

*p. 12, l. 9: It should be stressed that FAMOUS and MAR are consistently forced with the climate model output while RACMO is forced by reanalysis data. Also please specify the ERA forcing used and the period which was analysed.*

this has been done

*p. 14, Tab. 1: Please specify which ERA forcing is used here, Fettweis 2013 only has RACMO(ECMWF), and the RACMO(ERA) experiment that was used in this paper should be added to this table*

I'm actually confused by the notation in Fettweis 2013, which seems to me to use $RACMO_{ECMWF}$ and $RACMO_{ERA-Interim}$ in the same sense in places. Our Table 1 has been changed to use the same notation as the table in Fettweis 2013, but have used the same citation for ERA-Interim for both MAR(ERA_interim) and RACMO(ECMWF).

**Reviewer 2**

*You argue that Helsen et al., 2012 developed an empirical parameterization to translate global climate fields to usable boundary conditions. However, they state that their method is not usable for models where ablation areas are not resolved.*

this reference has been removed

*How does FAMOUS-ice prevent snow from growing infinitely thick? Does it include a firn-to-ice densification scheme, or do you cap the snow above a certain thickness?*

The snow scheme does include snowpack compaction and densification processes, although over time this alone would not prevent infinite columns of ice-density "snow" from forming where local accumulation is greater than ablation. In FAMOUS-ice we use the coupling scheme as outlined in section 8.1, whereby annual changes in snow mass are removed from FAMOUS every year and given to/taken from the mass of the ice sheet model via the SMB term. This is all clarified in the new paragraph on the evolution and relayering of the snowpack on page 9.

*For downscaling of the temperature and (incoming?) longwave radiation from the atmosphere to the elevation classes you use an empirically tuned method, where you assume that the near-surface climate gradients are similar to the climate gradients*

*in the lower atmosphere. Is this a valid assumption? For example, the temperature decreases with elevation along the GrIS near-surface, while the lower atmosphere has a temperature increase with height due to inversion.*

The surface temperature and downwelling longwave are downscaled with elevation using a fixed lapse rate. This is commonly done in this sort of downscaling (see references in the original paper), although it is undeniably a rather simplistic method and cannot reflect local conditions such as inversions that deviate from its cooling-with-height paradigm. The value used for this lapse rate is sometimes treated as a tunable parameter in order to achieve a realistic surface temperature distribution on the ice sheet. Used, as we do, to reflect subgrid-scale surface variation in a single land model gridbox, this near-surface-like scaling is more justified than trying to use the state of the free atmosphere column, especially in a lower-resolution model like FAMOUS where the vertical resolution in the atmosphere is rather low. There is some new clarification of our approach to this on page 6, line 14.

*I am not convinced you improve on the elevation class implementation analysed by Sellevold et al., 2019, as you are not comparing the same metric - that is, you do not subtract the grid-cell average from the same grid-box elevation class simulated value.*

We agree that the metric used in Sellevold et al. is more stringent than the vertical profiles shown in our original figures. We have now reproduced the Sellevold metric for selected downscaled fluxes on the ice sheet, and included it in the supplementary material (figure A6). This supports our claim that all the gradients act in the same sense in FAMOUS-ice as in the RCM used in Sellevold et al., although there are differences in the sensitivity of certain quantities to the elevation difference between each tile and the gridbox mean.

*Below 1000 m, the latent heat flux is positive in FAMOUS (Fig. 2), while negative in MAR and RACMO. First, it should be indicated whether the positive values means energy loss or gain at the surface. I am assuming the latter, as MAR and RACMO simulate climatological mean sublimation at lower elevations. Can you explain why FAMOUS simulates climatological mean deposition at these elevations?*

Thanks for this - there was a sign error in converting the FAMOUS-ice sublimation diagnostic into latent heat in this plot. Latent heat fluxes on Greenland in FAMOUS-ice are generally very small - this is noted in some additional discussion on page 8, line 2, and shown in the new supplementary figures.

*You argue that the shortwave down is too low because of a high biased cloud cover. I am surprised to see that this is not compensated for by an overestimation of longwave down. Is this due to the atmosphere being warmer in FAMOUS?*

Figure 2 now shows JJA longwave for more straightforward comparison with JJA shortwave, and some compensation can indeed be seen in the differences between FAMOUS-ice and MAR. There is no such longwave/shortwave compensation when comparing RACMO and FAMOUS-ice, but RACMO here is forced by the ERA-Interim reanalysis, so is subject to a very different background climate and this comparison is not necessarily to be expected.

*The simulation of albedo shows clear differences, particularly at higher elevations, when compared to MAR. You argue that the trigger for the lowered albedo at the higher elevations is a warm temperature bias at higher altitudes. Including a (supplementary) figure of April/May near-surface temperature or snow grain size compared to MAR/observations would likely strengthen this argument. Further, is it possible to assess whether the albedo is triggered too much and/or too early in the melt season?*

Further detailed analysis of different parts of the albedo parameterisation have shown that snow grain influences are not producing the widespread low summer albedos in these simulations. Instead, they come from the bare-ice values that are used when very dense firn appears at the surface, which seem to be used too readily. A similar feedback exists, with warmer temperatures lowering the albedo further via the bare-ice melt-pooling parameterisation and excessive melt potentially contributing to refreeze and increased densification of the snowpack underneath which may be exposed to the surface later. The change in attribution of the low albedo from snow-grain processes to the bare-ice parameterisation is noted in a new paragraph on page 13.

*Also, if the atmospheric temperatures are too high at high altitudes, the melt energy contribution from the sensible heat flux should also be higher. As you don't show the sensible heat flux, I am left to wonder how the simulation of the sensible heat flux is.*

Sensible heat fluxes are very low in FAMOUS-ice. This is noted now on page 8, line 2, and shown in a supplementary figure.

*How well is rain simulated over the ice sheet? As the ice sheet is warm at higher altitudes, is there an overestimation of rainfall at higher altitudes? Could this also contribute to a lower albedo?*

Rainfall distributions are now shown in a supplementary figure, and there is some discussion of precipitation bias in the Discussion on page 23, line 17. The model parameterisations mean that rain cannot affect the albedo calculation, as explained now on page 13, line 13.

*The presented surface mass balance shows a realistic distribution with ablation at lower elevations and accumulation*
5 *at higher elevations. However, the ablation areas are biased large and the high accumulation areas in the Southeast and Northwest show too little accumulation. As you don't present a transient climate simulation, I think the possible effect of these biases in transient simulations deserves a paragraph in the discussion section.*

Reviewer 1 also requested some consideration of how the model responds to climate change (See above) - there is now a new subsection (7.2) discussing SMB sensitivity, and number of supplementary figures showing the spatial distribution of changes
10 to key fields under climate change forcing.

*It would be easier to understand some of the thickness anomalies in Fig. 8 if you showed the ice velocities. Consider adding it to supplementary materials.*

this has been done, figure A15

*I found many occurrences of double parentheses with citations. If latex is used, use(text before citation ) or to avoid.*
15 these have been fixed

*P. 11, l. 12: replace fig:4 with figure 4.*

fixed

*P. 19, l. 2: "a model"*

fixed

20 *P. 19, l 2: "allow for coupled"*

I think this sentence is actually grammatically OK as it stands?

*Fig. 7, caption: replace x in: 10x km*

fixed. Thanks!

---

## Author Response (AR2)

**Response to review**

We would like to thank the reviewer again for their time and further comments, and are glad that they've found the revisions generally satisfactory. We will discuss their outstanding queries in more detail below, but will address two general themes first.

The remaining reviewer concerns seem to have been triggered by the new figure 8 in the revised manuscript showing the change in surface mass balance caused by the RCP4.5 climate forcing, and that the anomaly projected by FAMOUS does not have a similar pattern to the anomaly projected by the MAR RCM. We would first like to stress that what is being shown here is not the *absolute* SMB projected for the RCP4.5 forcing in each model, but the change compared to the 1980-1999 SMB base state shown in figure 6. This stress may be unnecessary, but it's not entirely clear from the review comments that they are talking about figure 8 as an anomaly plot, so we thought it best to be clear. Had we plotted the absolute SMBs for the RCP4.5 forcing for the two models (figure RR1) it would have much the same pattern as figure 6. Viewing the SMB in this absolute fashion shows a much more expected pattern of surface mass balance, but we do think it more useful to show the details of the anomaly compared to the baseline in terms of understanding how the climate of our model changes and the sensitivity of the SMB.

[Figure]

**Figure RR1.** Annual average surface mass balance (m/yr LWE) for the surface of Greenland under RCP4.5 in a) MAR (2080-2099 from a transient simulation forced by MIROC5 SSTs and atmospheric boundary conditions: minimum-4.02, maximum 1.98) and b) FAMOUS-ice (forced by constant 2080-2099 MIROC5 SSTs: minimum=-4.16; maximum=0.80). To visualise the distribution on sub-gridscale tiles, FAMOUS-ice results have been trilinearly mapped to the same topography as used in MAR. The green contour shows the MAR equilibrium line, the pink is the FAMOUS-ice equilibrium line.

We will proceed under the assumption that the reviewer did in fact recognise that figure 8 (and supplementary figures A12-A16) show only the changes of FAMOUS and MAR induced by RCP4.5 climate forcing compared to each model's late 20th century baseline, and that they are primarily unhappy that the physics response to climate change in FAMOUS do not closely match those simulated by MAR.

As stated previously main goal of this paper is a technical description, which we think worth publishing for its own sake. We agree that understanding the physics producing SMB and its sensitivity to climate change is important to understand, but it's not the primary aim of this paper - for FAMOUS-ice that is addressed in Gregory et al. (2020), and is being investigated further in a current project using large ensembles of parameter-perturbed model configurations. The climate results shown in our paper are from a single member of those ensembles, and whilst it is intended to be broadly representative of the behaviour of FAMOUS-ice it is not the only realisation of climate and climate change that the model can produce. We have tried to stress this more in the latest revision (eg page 7, line 29; page 14, line 9; page 24, line 18).

It is well-known that there is considerable uncertainty in the projection of future regional climate between models. Changes in clouds, and subsequently on radiative fluxes, make the single biggest contribution to that uncertainty, and since shortwave
forcing is such a significant influence on GrIS surface conditions, it is not surprising that two climate models might produce quite different projections of the sensitivity of surface shortwave and melt for the GrIS, even with similar local SST boundary conditions. FAMOUS is constrained only by SSTs and is free to develop its own large-scale atmospheric circulation, whilst MAR is additionally constrained by local MIROC5 atmospheric conditions. RCMs such as MAR may be considered the "gold-standard" for simulations of the recent climate for which they have been calibrated, but that's quite a different thing from
assuming that they will also have the correct cloud or climate sensitivity, something that is simply not known.

[Figure]

**Figure RR2.** Area integrated June-July-August average surface downwelling shortwave for GrIS from a range of CMIP5 and other models. Green bars: at the end of the 20th century from historical simulations; red bars: the anomaly between simulations with RCP4.5 greenhouse gas forcing and the historical

Figures RR2 and RR3 show the range of shortwave fluxes simulated for summer conditions in GrIS across a selection of climate models, mostly participants in CMIP5. The FAMOUS-ice example we have used in this paper is "FAMOUS-medium", and the MAR simulation we show in the paper for reference is labelled "MAR-MIROC5". The red anomaly bars (right axis) in RR2 shows that there is a wide variety of reductions in GrIS downwelling shortwave under RCP4.5 conditions across these
models, and whilst these configurations of FAMOUS-ice do have a relatively large reduction it is not the largest. MAR is at the very other end of the range, with a very small reduction in downwelling shortwave. There is no obvious relationship between the magnitude of the flux in the historical and the response to climate change. A similar plot of shortwave absorbed at the surface of GrIS shows an even wider range of responses, with some models absorbing more radiation in RCP4.5 than in the historical, and some less. FAMOUS is again near one extreme of this range and MAR near the other, but both have shortwave
responses that look reasonable when considering a broad range of climate models.

[Figure]

**Figure RR3.** Changes in June-July-August average surface downwelling shortwave (W/m-2) for GrIS from a range of CMIP5 and other models.

RR3 shows that patterns of shortwave radiation changes across these models also show a wide variety, and not just over Greenland. We have noted where the local radiative responses of FAMOUS-ice and MAR sit in the context of this broad range of model in the revised paper (page 19, line 1).

In general then, whilst we don't disagree with the need to understand the physics behind the sensitivity of climate processes
and surface mass balance in our model, we are satisfied that our illustrative example is within the range of behaviours generally displayed by other GCMs and RCMs, even if the pattern of SMB change in the example we used here do not match that in MAR. It may be instructive to investigate why that mismatch exists, but it is not of itself an indication that our model or our coupling schemes are not useful for their intended purpose. That being the case, and given that the primary aim of this paper is to describe the technical aspects of the FAMOUS and how it is coupled to the ice sheet model, we don't think that going into
significant extra detail of the climate behaviour of our example simulation in this GMD paper is justified - especially in the light of ongoing, large ensemble tuning work with this model that will be published in its own right when complete (noted on page 24, line 18).

We will respond now to individual aspects of the review. The reviewers' comments are in blue italics and our responses to each follow in black. Line/figure numbers given in our responses refer to the track-changed version of the resubmitted manuscript, line and figure numbers in reviewer comments refer to the revision of the discussion document that they reviewed.

**Individual comments**

*The new Fig. 8 exhibits a poor comparison to the SMB from the regional model MAR. Even though I am aware that model evaluation on the basis of other models is problematic, I would in this context consider MAR to be some kind of gold standard. The disagreement is substantial and I have doubts that the model is reliable, especially as FAMOUS was using the same ocean forcing as used in MAR. I am still not sure if here the climate model or the surface module is the problem, and this question should be answered.*

We've outlined our views on the reliability of model physics in FAMOUS compared to other models in the preamble above, and the current revision of the paper already contains a discussion of whether model biases are down to the background climate in FAMOUS or our approaches to downscaling and coupling. With regard to the particular pattern of the SMB anomaly simulated by the two models under climate change, we would note two additional points.

One is that there is additional variation in this pattern in FAMOUS with regard to the degree of climate change. In the paper we
used a MIROC5 RCP4.5 forcing since we had directly comparable simulation data available from both FAMOUS and MAR. Forcing our FAMOUS-ice configuration with stronger RCP8.5 SSTs (also from MIROC5) produces an SMB anomaly field with a stronger pattern of increased melt at the margins of the ice sheet (figure RR4,b), looking more like the MAR pattern (see also Fettweis et al., 2013) compared to the relatively ubiquitous melt anomaly in our FAMOUS-ice example under RCP4.5. The variation in regional change responses to climate change across models and scenarios is considerable. In general, the fixed
ice sheet mask of an atmosphere-only model like MAR can diagnose high negative SMBs at the ice sheet margin that may not be realistic, since the ice is modelled as a feature that can never disappear however much it is forced to melt. Although the ice extent changes in our illustrative simulation are not large at the point in the RCP4.5 simulation where we diagnosed the SMB anomaly in figure 8, it remains true that care must be taken interpreting details at the ice margins of models with different ice sheet masks.

We would further note that FAMOUS-ice is intended for long-term simulations of coupled climate-ice evolution. In an atmosphere-ice coupled model like FAMOUS-ice, the area integrated mass balance is often more important than its gradients, since ice dynamics will respond to determine the shape of the ice sheet. Table 1 shows that the integrated SMB component magnitudes are not so different from MAR's, and Gregory et al. (2020) show that the area-integrated magnitudes of SMB and its components in FAMOUS-ice are similar to RCMs over a wide range of 21st century climate forcings, so from that point of
view we are happy that our model is reliable and useful for its intended purpose.

[Figure]

**Figure RR4.** a) Surface mass balance (m/yr LWE) for the surface of Greenland for MIROC5 SSTs from years 2080-2099 under RCP8.5 in FAMOUS-ice b) Change in this SMB compared to the 1980-1999 baseline in figure 6

*If the climate model is the problem (it seems there is too much cloud cover), it would be helpful to also show the near surface temperatures. But neither in this manuscript nor in Gregory et al 2020 I have found an evaluation of the 2m air temperature distribution. (I take it that the model is generally slightly too warm over Greenland).*

We agree that surface air temperature would be a useful field to show, and have now re-run our FAMOUS-ice example to
produce this diagnostic. Since FAMOUS-ice uses an surface energy model to calculate snow melt the surface air temperature is not directly correlated with melt, of course. Surface air temperatures for the 1980-1999 reference period for MAR and our example FAMOUS-ice simulation are shown in new figure A7, with the response to RCP4.5 climate change in figure A12. A7 shows that FAMOUS is generally a little warmer than MAR, particularly at height. Since the low resolution of FAMOUS cannot support tight horizontal gradients this is not very surprising, although the specified lapse rate between the elevation
tiles will play a role here too. As is already shown, the shortwave radiation absorbed by GrIS in FAMOUS-ice and MAR is rather similar so this is probably not the reason for a warm surface bias. The climate change response in air temperature is quite different to MAR, which we would expect from differences in the response of the cloud and shortwave we've already talked about. In A12 MAR shows a general warming everywhere on the ice sheet, whilst FAMOUS-ice warms more strongly with height. This may indicate a change of atmospheric lapse rate in FAMOUS in the warmer climate (the lapse rate between
tiles is fixed), but this pattern is also correlated with the change in net shortwave absorbed at the surface. The net shortwave change is determined by the overlap between a region of reduction in albedo in the north and east, and the change in downward shortwave (figure A16,b) controlled by the cloud response over GrIS in FAMOUS-ice. Some discussion of this has been added to the revised paper (page 7, line 35; page 18, line 1).

*Furthermore I wonder if the disagreement in shortwave radiation is related to the coarse resolution rather than to a "wrong"*
*background climate (in that case downscaling of the atmospheric transmissivity might be considered). Is the shortwave radiation also biased outside of Greenland? How would Figure 2 look if the native FAMOUS resolution was used and MAR/RACMO was also remapped conservatively to the same coarse resolution?*

In development we didn't find that the downwelling shortwave varied with height in FAMOUS in a manner that suggested a significant scaling with tile elevation would be physically justified (already noted on page 7). FAMOUS data in figure 2
is plotted from the FAMOUS tiles rather than interpolated values and we are not convinced that we would learn we do not already know about the shortwave biases in FAMOUS by remapping and rebinning everything onto the coarse resolution grid of the FAMOUS atmosphere. Figure RR3 (using the native grid for all models, including FAMOUS) shows a wide variety of shortwave responses to climate change across CMIP5-era models, and we do not think that our FAMOUS example is uniquely different in any part of the northern hemisphere in this context. We consider analysing climate biases outside of GrIS beyond the scope of the current work.

*How does figure 2 look for the 2080-2099 period?*

[Figure]

**Figure RR5.** Profiles of surface exchange characteristics taken from every surface modelled on the Greenland ice sheet binned by surface elevation. Red: FAMOUS-ice; blue: MAR (2080-2099). FAMOUS-ice and MAR are forced by the MIROC5 climate (2080-2099). Latent heat is an annual average, negative values imply sublimation. Shading represents the full range of values found in each time-averaged elevation bin.

For reference here it is, although we feel that we have already discussed whether it is necessary to analyse further the differences in climate change response between FAMOUS-ice and MAR in the context of this paper, so it has not been added to the current revision. The variation of albedo with height matches better in the models for this period, the fundamental biases in downwelling long and shortwave remain. Since the albedoes match better, it thus follows that the shortwave absorbed will now differ more between the two models.

*In any case it is worrying that the albedo does not exhibit the typical spatial pattern. After tuning it is likely that albedo compensates biases in the climate, and in consequence the sensitivity of the surface scheme would be biased, too (good results for the wrong reasons) which could also explain the poor agreement in Fig. 8.*

It's not clear whether this comment refers to the albedo of the 20th century state in FAMOUS-ice or the albedo response under climate change. Assuming we're talking about the former (and referring to our responses above for whether we think agreement a necessary thing to achieve for what we show in figure 8) the paper does already discuss these matters. The albedo tuning is indeed a partial compensation for the downwelling shortwave bias, but it doesn't necessarily follow that the general climate sensitivity of the model will be biassed in a particular direction as a result of this nor, as we have seen above, that it causes the model to have a climate response that is exceptional either in pattern or magnitude in the context of a broader range of climate models.

The modern-day simulation of this FAMOUS-ice configuration does have a lower albedo at the margin of the GrIS than MAR (figure 4), so some of the margin is already near the lowest albedos that ice can realistically have. Under a moderate climate change scenario like RCP4.5 it is thus possible for MAR to simulate larger albedo reductions of the lowest-lying parts of the ice margin. This will undoubtedly contribute to differences in the pattern of SMB simulated in these models, although changes in downwelling shortwave and other energy components are also important. However, this difference in potential albedo response only affects the very edge of the ice and does not necessarily dominate the SMB response of a model over longer periods of time (when prolonged large negative SMB near the ice margin should remove those areas from the ice sheet and coupled flow dynamics become important) or under a stronger climate forcing which would affect larger areas of ice (figure RR3). A note discussing this has been added to the text (page 18, line 6).

*Finally I would like to point out that Kapsch et al, 2021 also present Greenland SMBs based on T31 resolution simulations, which seem to be fine; and from my own experience, PDD schemes are likewise able to produce a reasonable SMB response based on lapse rate corrected T31-temperatures (if insolation is similar to today's Greenland and temperatures are reasonable).*

I don't think we would dispute this, and would argue that we have shown FAMOUS-ice also produces reasonable simulations of present-day Greenland SMB, and can produce plausible future projections. The assumption made in PDD schemes of a climate- and geographically constant degree-day factor, even though it might give a reasonable SMB simulation for observed climate, introduces a potentially large and unquantifiable systematic uncertainty in modelling other climates, whereas an energy balance approach such as the one we (and Kapsch et al.) use should be more generally applicable.